



Hydrothermal activity lowers trophic diversity in Antarctic sedimented hydrothermal vents
James B. Bell[1,2], William D. K. Reid[3], David A. Pearce[4], Adrian G. Glover[2], Christopher J. Sweeting[5],
Jason Newton[6], & Clare Woulds[1]*
[1]School of Geography & Water@Leeds, University of Leeds, LS2 9JT, UK.
[2]Life Sciences Dept., Natural History Museum, Cromwell Rd, London SW7 5BD, UK
[3]Ridley Building, School of Biology, Newcastle University, NE1 7RU, UK
[4]Applied Sciences, Northumbria University, Newcastle, NE1 8ST, UK
[5]Ridley Building, School of Marine Science and Technology, Newcastle University, NE1 7RU, UK
[6]NERC Life Sciences Mass Spectrometry Facility, SUERC, East Kilbride G75 0QF, UK
* E-mail: c.woulds@leeds.ac.uk
Keywords: Stable Isotopes; Trophic Niche; Sedimented; Hydrothermal; Southern Ocean;
Microbial; 16S; PLFA





Abstract
Sedimented hydrothermal vents are those in which hydrothermal fluid is discharged through
sediments and are among the least studied deep-sea ecosystems. We present a combination of
microbial and biochemical data to assess trophodynamics between and within hydrothermally
active and off-vent areas of the Bransfield Strait (1050 – 1647m depth). Microbial composition,
biomass and fatty acid signatures varied widely between and within vent and non-vent sites and
provided evidence of diverse metabolic activity. Several species showed diverse feeding
strategies and occupied different trophic positions in vent and non-vent areas. Stable isotope
values of consumers were generally not consistent with feeding structure morphology. Niche
area and the diversity of microbial fatty acids reflected trends in species diversity and was
lowest at the most hydrothermally active site. Faunal utilisation of chemosynthetic activity was
relatively limited but was detected at both vent and non-vent sites as evidenced by carbon and
sulphur isotopic signatures, suggesting that hydrothermal activity can affect trophodynamics
over a much wider area than previously thought.



Section 1. Introduction
Sedimented hydrothermal vents (SHVs, a.k.a. Sediment-hosted hydrothermal vents), the
product of subsurface mixing between hydrothermal fluid and ambient seawater within the
sediment, are physically more similar to non-hydrothermal deep-sea habitats than they are to
high temperature, hard substratum vents (Bemis et al. 2012, Bernardino et al. 2012). This means
that, whilst they can host chemosynthetic obligate species, they can also be colonised by non-
specialist fauna, potentially offering an important metabolic resource in the nutrient-limited
deep-sea (Levin et al. 2009, Dowell et al. 2016). Sedimented vents have also been suggested to
act as evolutionary bridges between hard substratum vents and methane seeps (Kiel 2016). To
utilise in situ production at SHVs, fauna must overcome the environmental stress associated
with high-temperature, acidic and toxic conditions (Levin et al. 2013, Gollner et al. 2015). The
combination of elevated toxicity and in-situ organic matter (OM) production results in a
different complement of ecological niches between vents and background conditions that elicits
compositional changes along a productivity-toxicity gradient (Bernardino et al. 2012, Gollner et
al. 2015, Bell et al. 2016b). Hydrothermal sediments offer different relative abundances of
chemosynthetic and photosynthetic organic matter, depending upon supply of surface-derived
primary productivity, which may vary with depth and latitude, and levels of hydrothermal
activity (Tarasov et al. 2005). In shallow environments (<200 m depth), where production of
chemosynthetic and photosynthetic organic matter sources can co-occur, consumption may still
favour photosynthetic OM over chemosynthetic OM as this does not require adaptions to
environmental toxicity (Kharlamenko et al. 1995, Tarasov et al. 2005, Sellanes et al. 2011). The
limited data available concerning trophodynamics at deep-sea SHVs, from the Arctic, indicate
that diet composition varies widely between taxa, ranging between 0 – 87 % contribution from



chemosynthetic OM (Sweetman et al. 2013). Thus, understanding of the significance of
chemosynthetic activity in these settings is very limited.

Sedimented hydrothermal vents host diverse microbial communities (Teske et al. 2002,
Kallmeyer & Boetius 2004). Microbial communities are a vital intermediate between
hydrothermal fluid and metazoan consumers, and thus their composition and isotopic
signatures are of direct relevance to metazoan food webs. The heat flux associated with
hydrothermal activity provides thermodynamic benefits and constraints to microbial
communities (Kallmeyer & Boetius 2004, Teske et al. 2014) whilst accelerating the degradation
of organic matter, giving rise to a wide variety of compounds including hydrocarbons and
organic acids (Martens 1990, Whiticar & Suess 1990, Dowell et al. 2016). Microbial aggregations
are commonly visible on the sediment surface at SHVs (Levin et al. 2009, Sweetman et al. 2013,
Dowell et al. 2016). However, active communities are also distributed throughout the underlying
sediment layers, occupying a wide range of geochemical and thermal niches (reviewed by Teske
et al. 2014). This zonation in microbial function and composition is very strong and has been
extensively studied in Guaymas basin hydrothermal sediments. Sedimented chemosynthetic
ecosystems may present several sources of organic matter to consumers (Bernardino et al. 2012,
Sweetman et al. 2013, Yamanaka et al. 2015) and the diverse microbial assemblages can support
a variety of reaction pathways, including methane oxidation, sulphide oxidation, sulphate
reduction and nitrogen fixation (Teske et al. 2002, Dekas et al. 2009, Jaeschke et al. 2014).
Phospholipid fatty acid (PLFA) analysis can be used to describe recent microbial activity and
$\delta^{13}C$ signatures (Boschker & Middelburg 2002, Yamanaka & Sakata 2004, Colaço et al. 2007).
Although it can be difficult to ascribe a PLFA to a specific microbial group or process, high
relative abundances of certain PLFAs can be strongly indicative of chemoautotrophy (Yamanaka



& Sakata 2004, Colaço et al. 2007), and can support an understanding of microbial ecosystem
function in hydrothermal sediments (e.g. in western pacific vents, see Yamanaka & Sakata 2004).

Macrofaunal assemblages of the Bransfield SHVs were strongly influenced by hydrothermal
activity (Bell et al. 2016b). Bacterial mats were widespread across Hook Ridge, where variable
levels of hydrothermal activity were detected (Aquilina et al. 2013). Populations of siboglinid
polychaetes (*Sclerolinum contortum* and *Siboglinum* sp.), were found at Hook Ridge and non-
hydrothermally active sites (Sahling et al. 2005, Georgieva et al. 2015, Bell et al. 2016b). These
species are known to harbour chemoautotrophic endosymbionts (Schmaljohann et al. 1990,
Eichinger et al. 2013, Rodrigues et al. 2013). Stable isotope analysis (SIA) is a powerful tool to
assess spatial and temporal patterns in faunal feeding behaviour and has been used to study
trophodynamics and resource partitioning in other SHVs, predominately in the Pacific (Fry et al.
1991, Levin et al. 2009, Portail et al. 2016). Stable isotopic analyses provide inferential measures
of different synthesis pathways and can elucidate a wide range of autotrophic or feeding
behaviours. Carbon and sulphur isotopes are used here to delineate food sources and nitrogen
is used as a measure of trophic position. The signature of source isotope ratios ($\delta^{13}$C & $\delta^{34}$S) is
influenced by the isotopic ratio of the chemical substrate, and the fractionation associated with
the metabolic process involved and thus, different fixation pathways elicit different isotopic
signatures, even when they utilise the same source (e.g. DIC) (Fry et al. 1991). Possible $\delta^{13}$C
isotopic values of sources in the Bransfield Strait include: ∼-40 ‰ for thermogenic methane; ∼-
27 ‰ for suspended particulate matter or ∼-15 ‰ for ice algae (Whiticar & Suess 1990, Mincks
et al. 2008, Henley et al. 2012, Young et al. 2013). As an example, *Siboglinum* spp. can use a range
of resources, including methane or dissolved organic matter (Southward et al. 1979,
Schmaljohann et al. 1990, Thornhill et al. 2008, Rodrigues et al. 2013), making SIA an ideal way
in which to examine resource utilisation in these settings (Levin et al. 2009, Soto 2009). We also



apply the concept of an isotopic niche (Layman et al. 2007) whereby species or community
trophic activity is inferred from the distribution of stable isotopic data in two or three
dimensional isotope space.

Hypotheses

We used a combination of microbial diversity data based sequencing and compound specific
isotopic analyses and bulk isotopic data from sediment, microbial, macro- and megafaunal
samples to investigate resource utilisation, niche partitioning and trophic structure at vent and
background sites in the Bransfield Strait to test the following hypotheses: 1) Siboglinid species
subsist upon chemosynthetically-derived OM; 2) Chemosynthetic organic matter will be a
significant food source at SHVs; 3) Stable isotope signatures will reflect a-priori functional
designations defined by faunal morphology and 4) Fauna will have distinct niches between vents
and background areas.



Section 2. Materials and Methods

2.1. Sites and Sampling

Samples were collected; during RRS *James Cook* cruise JC55 in the austral summer of 2011 (Tyler
et al. 2011), from three raised edifices along the basin axis (Hook Ridge, the Three Sisters and
The Axe) and one off-axis site in the Bransfield Strait (1024 – 1311m depth; Fig. 1; Table 1). We
visited two sites of variable hydrothermal activity (Hook Ridge 1 and 2) and three sites where
hydrothermal activity was not detected (Three Sisters, the Axe and an Off-Axis site) (Aquilina et
al. 2013). Of the two hydrothermal sites, Hook Ridge 2 was had higher hydrothermal fluid
advection rates and pore fluid temperature but lower concentrations of sulphide and methane
(Dählmann et al. 2001, Aquilina et al. 2013, Aquilina et al. 2014).

Samples were collected with a series of megacore deployments, using a Bowers & Connelly
dampened megacorer (1024 – 1311 m depth) and a single Agassiz trawl at Hook Ridge (1647 m
depth). With the exception of salps, all microbial and faunal samples presented here were from
megacore deployments. For a detailed description of the megacore sampling programme and
macrofaunal communities, see Bell et al. (2016b). Sampling consisted of 1 – 6 megacore
deployments per site, with 2 – 5 tubes pooled per deployment (Bell et al. 2016b). Cores were
sliced into 0 – 5 cm and 5 – 10 cm partitions and macrofauna were retained on a 300μm sieve.
Residues were preserved in either 80 % ethanol or 10 % buffered formalin initially and then
stored in 80% ethanol after sorting (Bell et al. 2016b). Fauna were sorted to species/
morphospecies level (for annelid and bivalve taxa); family level (for peracarids) and higher
levels for less abundant phyla (e.g. echiurans). Salps were collected using an Agassiz trawl and
samples were immediately picked and frozen at -80 ℃ and subsequently freeze-dried.




2.2. Microbiology Sequencing

Samples of surface sediment (0 – 1 cm below seafloor (cmbsf)) were taken from megacores the
two Hook Ridge sites and the off-axis site and frozen (-80℃). DNA was extracted from the
sediment by Mr DNA (Shallowater, TX, USA) using an in-house standard 454 pipeline. The
resultant sequences were trimmed and sorted using default methods in Geneious (v.9.1.5 with
RDP v.2.8 and Krona v.2.0) and analysed in the Geneious '16 Biodiversity Tool'
(https://16s.geneious.com/16s/help.html; (Wang et al. 2007, Ondov et al. 2011, Biomatters

154    2014).


2.3. Phospholipid Fatty Acids

Samples of 3 – 3.5 g of freeze-dried sediment from Hook Ridge 1 & 2, the off-vent site and the
Three Sisters were analysed at the James Hutton Institute (Aberdeen, UK) following the
procedure detailed in Main et al. (2015), which we summarise below. Samples were from the
top 1 cm of sediment for all sites except Hook Ridge 2 where sediment was pooled from two core
slices (0 – 2 cm), due to sample mass limitations. Lipids were extracted following a method
adapted from Bligh (1959), using a single phase mixture of chloroform: methanol: citrate buffer
(1:2:0.8 v-v:v). Lipids were fractionated using 6 ml ISOLUTE SI SPE columns, preconditioned
with 5 ml chloroform. Freeze-dried material was taken up in 400 μL of chloroform; vortex mixed
twice and allowed to pass through the column. Columns were washed in chloroform and acetone
(eluates discarded) and finally 10 ml of methanol. Eluates were collected, allowed to evaporate
under a $N_2$ atmosphere and frozen (-20 ℃).



PLFAs were derivitised with methanol and KOH to produce fatty acid methyl esters (FAMEs).
Samples were taken up in 1 mL of 1:1 (v:v) mixture of methanol and toluene. 1 mL of 0.2 M KOH
(in methanol) was added with a known quantity of the C19 internal standard (nonadecanoic
acid), vortex mixed and incubated at 37 ℃ for 15 min. After cooling to room temperature, 2 mL
of isohexane:chloroform (4:1 v:v), 0.3 mL of 1 M acetic acid and 2 mL of deionized water was
added to each vial. The solution was mixed and centrifuged and the organic phase transferred to
a new vial and the remaining aqueous phase was mixed and centrifuged again to further extract
the organic phase, which was combined with the previous. The organic phases were evaporated
under a $N_2$ atmosphere and frozen at -20 ℃.

Samples were taken up in isohexane to perform gas chromatography-combustion-isotope ratio
mass spectrometry (GC-C-IRMS). The quantity and $\delta^{13}C$ values of individual FAMEs were
determined using a GC Trace Ultra with combustion column attached via a GC Combustion III to
a Delta V Advantage isotope ratio mass spectrometer (Thermo Finnigan, Bremen). The $\delta^{13}C_{VPDB}$
values (‰) of each FAME were calculated with respect to a reference gas of $CO_2$, traceable to
IAEA reference material NBS 19 TS-Limestone. Measurement of the Indiana University reference
material hexadecanoicacid methyl ester (certified $\delta^{13}C_{VPDB}$ -30.74 ± 0.01‰) gave a value of
30.91 ± 0.31‰ (mean ± s. d., n = 51). Combined areas of all mass peaks (m/z 44, 45 and 46),
following background correction, were collected for each FAME. These areas, relative to the
internal C19:0 standard, were used to quantify the 34 most abundant FAMEs and related to the
PLFAs from which they are derived (Thornton et al. 2011).

Bacterial biomass was calculated using transfer functions from the total mass of four PLFAs
(i14:0, i15:0, a15:0 and i16:0), estimated at 14 % of total bacterial PLFA, which in turn is
estimated at 5.6 % of total bacterial biomass (Boschker & Middelburg 2002).




2.4. Bulk Stable Isotopes

All bulk isotopic analyses were completed at the East Kilbride Node of the Natural Environment
Research Council Life Sciences Mass Spectrometry Facility. Specimens with carbonate structures
(e.g. bivalves) were physically decarbonated and all specimens were rinsed in de-ionised water
(e.g. to remove soluble precipitates such as sulphates) and cleaned of attached sediment before
drying. Specimens dried for at least 24 hours at 50°C and weighed (mg, correct to 3 d.p.) into tin
capsules and stored in a desiccator whilst awaiting SIA. Samples were analysed by continuous
flow isotope ratio mass spectrometer using a Vario-Pyro Cube elemental analyser (Elementar),
coupled with a Delta Plus XP isotope ratio mass spectrometer (Thermo Electron). Each of the
runs of CN and CNS isotope analyses used laboratory standards (Gelatine and two amino acid-
gelatine mixtures) as well as the international standard USGS40 (glutamic acid). CNS
measurements used the internal standards (MSAG2: (Methanesulfonamide/ Gelatine and M1:
Methionine) and the international silver sulphide standards IAEA-S1, S2 and S3. All sample runs
included samples of freeze-dried, powdered *Antimora rostrata* (ANR), an external reference
material used in other studies of chemosynthetic ecosystems (Reid et al. 2013, Bell et al. 2016a),
used to monitor variation between runs and instruments (supplementary file 1). Instrument
precision (S.D.) for each isotope measured from ANR was 0.42 ‰, 0.33 ‰ and 0.54 ‰ for
carbon, nitrogen and sulphur respectively. The reference samples were generally consistent
except in one of the CNS runs, which showed unusual $\delta^{15}N$ measurements (S1), so faunal $\delta^{15}N$
measurements from this run were excluded as a precaution. Stable isotope ratios are all reported
in delta ($\delta$) per mil (‰) notation, relative to international standards: V-PDB ($\delta^{13}C$); Air ($\delta^{15}N$)
and V-CDT ($\delta^{34}S$). Machine error, relative to these standards ranged 0.01 – 0.23 for $\delta^{13}C$, for 0.01
– 0.13 $\delta^{15}N$ and 0.13 – 3.04 for $\delta^{34}S$. One of the Sulphur standards (Ag$_2$S IAEA: S2) had a notable



difference from the agreed measurements, suggesting either a compromised standard or poor
instrument precision. This error was not observed in other standards, or the reference material
used, but given the uncertainty here; only $\delta^{34}S$ differences greater than 3 ‰ are considered as
being significant.

A combination of dual- ($\delta^{13}C$ & $\delta^{15}N$, 319 samples) and tri-isotope ($\delta^{13}C$, $\delta^{15}N$ & $\delta^{34}S$, 83 samples)
techniques was used to describe bulk isotopic signatures of 43 species of macrofauna (35 from
non-vent sites, 19 from vent sites and 11 from both), 3 megafaunal taxa and sources of organic
matter. Samples submitted for carbon and nitrogen (CN) analyses were pooled if necessary to
achieve an optimal mass of 0.7 mg (± 0.5 mg). Where possible, individual specimens were kept
separate in order to preserve variance structure within populations but in some cases, low
sample mass meant individuals had to be pooled (from individuals found in replicate
deployments). Optimal mass for Carbon-Nitrogen-Sulphur (CNS) measurements was 2.5 mg
(±0.5 mg) and, as with CN analyses, specimens were submitted as individual samples or pooled
where necessary. Samples of freeze-dried sediment from each site were also submitted for CNS
analyses (untreated for NS and acidified with 6M HCl for C). Acidification was carried out by
repeated washing with acid and de-ionised water.

Specimens were not acidified. A pilot study, and subsequent results presented here, confirmed
that the range in $\delta^{13}C$ measurements between acidified (0.1M and 1.0M HCl) was within the
untreated population range, in both polychaetes and peracarids and that acidification did not
notably or consistently reduce $\delta^{13}C$ standard deviation (Table 2). In the absence of a large or
consistent treatment effect, the low sample mass, (particularly for CNS samples) was dedicated
to increasing replication and preserving integrity of $\delta^{15}N$ & $\delta^{34}S$ measurements instead of
separating carbon and nitrogen/ sulphur samples (Connolly & Schlacher 2013).





Formalin and ethanol preservation effects can both influence the isotopic signature of a sample
(Fanelli et al. 2010, Rennie et al. 2012). Taxa that had several samples of each preservation
method from a single site (to minimise intra-specific differences) were examined to determine
the extent of isotopic shifts associated with preservation effects. Carbon and nitrogen isotopic
differences between ethanol and formalin preserved samples ranged between 0.1 ‰ – 1.4 ‰
and 0.4 ‰ – 2.0 ‰ respectively. Differences across all samples were not significant (Paired t-
test, $\delta^{13}$C: t = 2.10, df = 3, p = 0.126 and $\delta^{15}$N: t=1.14, df = 3, p = 0.337). Given the unpredictable
response of isotopic signatures to preservation effects (which also cannot be extricated from
within-site, intraspecific variation) it was not possible to correct isotopic data (Bell et al. 2016a).
This contributed an unavoidable, but generally quite small, source of error in these
measurements.

2.5. Statistical Analyses

All analyses were completed in the R statistical environment (R Core Team 2013). Carbon and
nitrogen stable isotopic measurements were divided into those from vent or non-vent sites and
averaged by taxa and used to construct a Euclidean distance matrix (Valls et al. 2014). This
matrix was used to conduct a similarity profile routine (SIMPROF, 10 000 permutations, p = 0.05,
Ward linkage) using the clustsig package (v1.0) (Clarke et al. 2008, Whitaker & Christmann
2013) to test for significant structure within the matrix. The resulting cluster assignments were
compared to a-priori feeding groups (Bell et al. 2016b) using a Spearman Correlation Test (with
9 999 Monte Carlo resamplings) using the coin package (v1.0-24) (Hothorn et al. 2015). Isotopic
signatures of species sampled from both vent and non-vent sites were also compared with a one-
way ANOVA with Tukey's HSD pairwise comparisons (following a Shapiro-Wilk normality test).




Mean faunal measurements of $\delta^{13}$C & $\delta^{15}$N were used to calculate Layman metrics for each site
(Layman et al. 2007), sample-size corrected standard elliptical area (SEAc) and Bayesian
posterior draws (SEA.B, mean of $10^5$ draws ± 95 % credibility interval) in the SIAR package
(v4.2) (Parnell et al. 2010, Jackson et al. 2011). Differences in SEA.B between sites were
compared in mixSIAR. The value of p given is the proportion of ellipses from group A that were
smaller in area than those from group B (e.g. if p = 0.02, then 2 % of posterior draws from group
A were smaller than the group B mean) and is considered to be a semi-quantitative measure of
difference in means (Jackson et al. 2011).





Section 3. Results

3.1. Differences in microbial composition along a hydrothermal gradient

A total of 28,767, 35,490 and 47,870 sequences were obtained from the off-axis site and the vent
sites, Hook Ridge 1 and 2, respectively. Bacteria comprised almost the entirety of each sample,
with Archaea being detected only in the Hook Ridge 2 sample (< 0.1 % of sequences). Hook Ridge
1 was qualitatively more similar to the off-axis site than Hook Ridge 2. Both Hook Ridge 1 (vent)
and the off-vent site, BOV (non-vent), were dominated by Proteobacteria (48 % and 61 % of
reads respectively; Fig. 2), whereas Flavobacteriia dominated Hook Ridge 2 (43 %, 7 – 12 %
elsewhere) with Proteobacteria accounting for a smaller percentage of sequences (36 %; Fig. 2).
By sequence abundance, Flavobacteriia were the most clearly disparate group between Hook
Ridge 2 and the other sites. Flavobacteriia were comprised of 73 genera at Hook Ridge 2, 60
genera at BOV and 63 genera at HR1, of which 54 genera were shared between all sites. Hook
Ridge 2 had 15 unique flavobacteriial genera but these collectively accounted for just 0.9% of
reads, indicating that compositional differences were mainly driven by relative abundance,
rather than taxonomic richness.

The most abundant genus from each site was *Arenicella* at BOV and HR1 (7.1 and 5.2 % of reads
respectively) and *Aestuariicola* at HR2 (6.9 % of reads). The four most abundant genera at both
BOV and HR1 were *Arenicella* (γ-proteobacteria), *Methylohalomonas* (γ-proteobacteria),
*Pasteuria* (Bacilli) & *Blastopirellula* (Planctomycetacia), though not in the same order, and
accounted for 17.2% and 16.0 % of reads respectively. The four most abundant genera at HR2,
accounting for 20.2 % of reads were *Aestuariicola*, *Lutimonas, Maritimimonas* & *Winogradskyella*



(all Flavobacteriia). The genera *Arenicella* and *Pasteuria* were the most relatively abundant
across all sites (2.2 % – 7.1 % and 1.7 % – 5.0 % of reads respectively).

3.2. Microbial fatty acids

A total of 37 sedimentary PLFAs were identified across all sites, in individual abundances
ranging between 0 % – 26.4 % of total PLFA (Table 3; Supplementary Fig 1). All lipid samples
were dominated by saturated and mono-unsaturated fatty acids (SFAs and MUFAs), comprising
91 % – 94 % of PLFA abundance per site. The most abundant PLFAs at each site were
16:0(15.7 % – 26.4 %), 16:1ω7c (11.5 % – 20.0 %) and 18:1ω7 (4.8 % – 16.9 %; Table 3). PLFA
profiles from each of the non-vent sites sampled (Off-axis and the Three Sisters, 33 and 34 PLFAs
respectively) were quite similar (Table 3) and shared all but one compound (16:1ω11c, present
only at the non-vent Three Sisters site). Fewer PLFAs were enumerated from Hook Ridge 1 and
2 (31 and 23 respectively), including 3 PLFAs not observed at the non-vent sites (br17:0, 10-Me-
17:0 & 10-Me-18:0), which accounted for 0.5 % – 1.2 % of the total at these sites. Poly-
unsaturated algal biomarkers (20:5ω3 and 22:6 ω3) were only detected at the non-vent site
(0.83 – 1.57 % of total FA abundance). Hook Ridge 2 had the lowest number of PLFAs and the
lowest total PLFA biomass of any site, though this was due in part to the fact that this sample
had to be pooled from the top 2 cm of sediment (top 1cm at other sites). Bacterial biomass was
highest at Hook Ridge 1 and ranged 85 mg C m$^{-2}$ – 535 mg C m$^{-2}$ (Table 3).

PLFA carbon isotopic signatures ranged -56 ‰ to -20 ‰ at non-vent sites and -42 ‰ to -8 ‰
at vent sites (Table 3). Weighted average δ13C values were quite similar between the non-vent
sites and Hook Ridge 1 (-30.5 ‰ and -30.1 ‰ respectively), but were heavier at Hook Ridge 2
(-26.9 ‰; Table 3). Several of the PLFAs identified had a large range in δ13C between samples

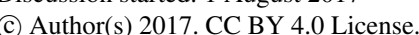



(including 16:1ω11t $\delta^{13}$C range = 17.2 ‰ or 19:1ω8 $\delta^{13}$C range = 19.1 ‰), even between the
non-vent sites (e.g. 18:2ω6, 9, $\Delta\delta^{13}$C = 24.4; Table 3). Of the 37 PLFAs, 7 had a $\delta^{13}$C range of >
10 ‰ but these were comparatively minor and individually accounted for 0 % – 4.9 % of total
abundance. Average $\delta^{13}$C range was 6.3 ‰ and a further 11 PLFAs had a $\delta^{13}$C range of > 5 ‰,
including some of the more abundant PLFAs, accounting for 36.8 ‰ – 46.6 % at each site. PLFAs
with small $\delta^{13}$C ranges (< 5 ‰) accounted for 44.6 % – 54.4 % of total abundance at each site.

3.3. Description of bulk isotopic signatures

Most faunal isotopic signatures were within a comparatively narrow range ($\delta^{13}$C: -30 ‰ to -
20 ‰, $\delta^{15}$N: 5 ‰ to 15 ‰ and $\delta^{34}$S: 10 ‰ to 20 ‰) and more depleted isotopic signatures
were usually attributable to siboglinid species (Fig. 3). *Siboglinum* sp. (found at all non-vent
sites) had mean $\delta^{13}$C and $\delta^{15}$N values of -41.4 ‰ and -8.9 ‰ respectively and *Sclerolinum*
*contortum* (predominately from Hook Ridge 1 but found at both vent sites) had values of -
20.5 ‰ and -5.3‰ respectively. Some non-endosymbiont bearing taxa (e.g. macrofaunal
neotanaids from the off-axis site and megafaunal ophiuroids at Hook Ridge 2) also had notably
depleted $\delta^{15}$N signatures (means -3.6‰ to 2.6 ‰ respectively; Fig. 3).

Isotopic signatures of sediment organic matter were similar between vents and non-vents for
$\delta^{13}$C and $\delta^{15}$N but $\delta^{34}$S was significantly greater at non-vent sites ($p < 0.05$, Table 4; Fig. 4).
Variability was higher in vent sediments for all isotopic signatures. Faunal isotopic signatures
for $\delta^{13}$C and $\delta^{34}$S ranged much more widely than sediment signatures and indicate that sediment
organics were a mixture of two or more sources of organic matter. A few macrofaunal species
had relatively heavy $\delta^{13}$C signatures that exceeded -20 ‰ that suggested either a heavy source
of carbon or marine carbonate in residual exoskeletal tissue, particularly for peracarids (~0 ‰).



Samples of pelagic salps from Hook Ridge had mean values for $\delta^{13}C$ of -27.4 ‰ (± 0.9) and $\delta^{34}S$
of 21.5 ‰ (± 0.8).

3.4. Comparing macrofaunal morphology and stable isotopic signatures

Averaged species isotopic data were each assigned to one of four clusters (SIMPROF, p = 0.05;
Supplementary Figure 3). No significant correlation between a-priori (based on morphology)
and a-posteriori clusters (based on isotopic data) was detected (Spearman Correlation Test: Z =
-1.34; N = 43; p = 0.18). Clusters were mainly discriminated based on $\delta^{15}N$ values and peracarids
were the only taxa to be represented in all of the clusters, indicating high trophic diversity.

Several taxa found at both vent and non-vent sites were assigned to different clusters between
sites. A total of eleven taxa were sampled from both vent and non-vent regions, of which four
were assigned to different clusters at vent and non-vent sites. Neotanaids (Peracarida:
Tanaidacea) had the greatest Euclidean distance between vent/ non-vent samples (11.36),
demonstrating clear differences in dietary composition (Fig. 5). All other species were separated
by much smaller distances between regions (range: 0.24 to 2.69). Raw $\delta^{13}C$ and $\delta^{15}N$ values were
also compared between vent and non-vent samples for each species (one-way ANOVA with
Tukey HSD pairwise comparisons). Analysis of the raw data indicated that $\delta^{13}C$ signatures were
different for neotanaids only and $\delta^{15}N$ were different for neotanaids and an oligochaete species
(*Limnodriloides* sp.) (ANOVA, p < 0.01, Fig. 5).

3.5.Community-level trophic metrics



All site niches overlapped (mean = 50 %, range = 30 – 82 %) and the positions of ellipse centroids
were broadly similar for all sites (Table 5; Fig 6). Vent site ellipse areas were similar but
significantly smaller than non-vent ellipses (SEA.B, n = $10^5$, p = < 0.05). There were no significant
differences in ellipse area between any of the non-vent sites. Ranges in carbon sources (dCr)
were higher for non-vent sites (Table 5) indicating a greater trophic diversity in background
conditions. Nitrogen range (dNr, Table 5) was similar between vents and non-vents suggesting
a similar number of trophic levels within each assemblage. All site ellipses had broadly similar
eccentricity (degree of extension along long axis), ranging 0.85 – 0.97 (Table 5), however theta
(angle of long axis) differed between vent and non-vent sites (-1.43 to 1.55 at Hook Ridge, 0.67
to 0.86 at non-vent sites). Range in nitrogen sources was more influential at vent sites as
*Sclerolinum contortum*, which had very low $\delta^{15}$N signatures but similar $\delta^{13}$C values, when
compared with non-endosymbiont bearing taxa from the same sites. The strongly depleted $\delta^{13}$C
measurements of *Siboglinum* sp. meant that ellipse theta was skewed more towards horizontal
(closer to zero) for non-vent sites.



Section 4. Discussion

4.1. Microbial signatures of hydrothermal activity

PLFA profiles between the off-axis site and the Three Sisters indicated similar bacterial biomass
at each of these non-vent sites and that bacterial biomass varied much more widely at Hook
Ridge (Table 3). The Hook Ridge 2 sample is not directly comparable to the others as it was
sampled from sediment 0 – 2 cmbsf (owing to sample mass availability), though organic carbon
content, hydrogen sulphide flux and taxonomic diversity were all lower at this site and may
support suggestion of a lower overall bacterial biomass (Aquilina et al. 2013, Bell et al. 2016b).
The very high bacterial biomass at Hook Ridge 1 suggests a potentially very active bacterial
community, comparable to other hydrothermal sediments (Yamanaka & Sakata 2004) but
$\delta 13C_{org}$ was qualitatively similar to non-vent sites, implying that chemosynthetic activity was
comparatively limited, or that the isotopic signatures of the basal carbon source (e.g. DIC) and
the fractionation associated with FA synthesis resulted in similar $\delta^{13}C$ signatures.

Hook Ridge 1 PLFA composition was intermediate between non-vent sites and Hook Ridge 2
(Supplementary Fig. 2) but the PLFA suite was quite similar between Hook Ridge 1 and the off-
axis site (Fig. 2). A small number of the more abundant PLFAs had notable differences in relative
abundance between vent/ non-vent sites (Table 3). For example, 16:1ω7, which has been linked
to sulphur cycling pathways (Colaço et al. 2007) comprised 14.0 % – 15.2 % of abundance at
non-vent sites and 20.0 % – 23.5 % at vent sites. However, 18:1ω7, also a suggested PLFA linked
to thio-oxidation (McCaffrey et al. 1989, Colaço et al. 2007) occurred in lower abundance at vent
sites (4.8 % – 11.1 %) than non-vent sites (15.9 % – 16.9 %), and was also abundant in deeper
areas of the Antarctic shelf (Würzberg et al. 2011). These results further suggest that

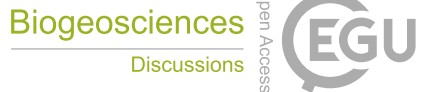

chemosynthetic activity was relatively limited since, although there were differences between
sites in PLFAs that are potentially indicative of chemosynthetic activity, these were not
necessarily consistent between different PLFAs. The metabolic provenance of several of the
more abundant PLFAs is also still uncertain. A number of fatty acids have been linked, though
not exclusively, to chemoautotrophy, such as 10-Me-16:0 (*Desulfobacter* or *Desulfocurvus,*
Sulphate reducers) and 18:1ω7 (Yamanaka & Sakata 2004, Colaço et al. 2007, Klouche et al. 2009,
Boschker et al. 2014) and the presence of these FAs may be consistent with the hydrothermal
signature of the sediment microbial community. Together C16:1ω7c and C18:1ω7 accounted for
~25-35% of the total PLFA suite. While they can be more generally associated with gram-
negative eubacteria, these PLFAs in sediment samples have frequently been linked to sulphur
oxidising bacteria (Pond et al. 1998, Yamanaka & Sakata 2004, Boschker et al. 2014). Their
dominance of the suite in the Bransfield Strait is similar to sediments from a vent in the Barbados
Trench, where together C16:1ω7 and C18:1ω7 contributed up to 50% of PLFAs (Guezennec &
Fiala-Medioni 1996). They have also been shown to be dominant in the PLFA suites of sulphur
oxidising bacteria such as *Beggiatoa* (e.g. Guezennec et al. 1998). The PLFA suites also contained
notable proportions of compounds normally associated with sulphate reducing bacteria
(Kohring et al. 1994, Boschker et al. 2014). These included iC15:0, aiC15:0, 1C17:0 and aiC17:0,
which together constituted ~8-12 % of the PLFA suite. In addition, C16:1ω5c was relatively
abundant (Supplementary figure 1), and minor amounts of 10MeC16:0, C17:1ω8c, and
cycloC17:0 were present. These have also been used as indicators of sulphate reducing bacteria,
and sometimes of particular groups (e.g. Guezennec & Fiala-Medioni 1996, Boschker et al. 2014).
These compounds indicate the presence of sulphate reducing bacteria, although perhaps not as
the dominant group. Although the PLFA suite was indicative of active sulphur cycling activity, it
remains difficult to be conclusive about the origin of most PLFAs even those which have been




regularly observed in chemosynthetic contexts (e.g. 18:1ω7) may still be abundant elsewhere
(Würzberg et al. 2011).

Unsurprisingly, long chain fatty acids (>C22) indicative of land plants (e.g. Yamanaka & Sakata
2004) were negligible or absent. More notably, the typical indicators of marine phytoplankton
production C20:3ω5 and C22:6ω3 were very minor constituents, never accounting for more
than 3% of PLFAs and only detected at the non-vent sites Off Vent and Middle Sister. While their
low abundance is at least partially accounted for by rapid degradation of polyunsaturated fatty
acids during sinking through the water column (Veuger et al. 2012), it also suggests that
sedimentary PLFAs are dominantly of bacterial origin, whether that be due to bacterial
reworking of photosynthetic organic matter, or in situ production, and that this influence of
bacterial activity is greater at vent sites than at non-vent sites.

Heavier carbon isotopic signatures (> -15 ‰) are generally associated with rTCA cycle carbon
fixation (Hayes 2001, Hugler & Sievert 2011, Reid et al. 2013), suggesting that this pathway may
have been active at the vent sites, albeit at probably quite low rates. Conversely, many of the
lightest $\delta^{13}$C signatures (e.g. 19:1ω8, -56.6 ‰, off-axis site) were associated with the non-vent
sites, however 19:1ω8 has not been directly associated with a particular bacterial process
(Koranda et al. 2013, Dong et al. 2015). Lower PLFA carbon isotope signatures with small ranges
(e.g. -60 ‰ to -50 ‰) could also be indicative of methane cycling, but most PLFAs at all sites
had $\delta^{13}$C of > -40 ‰.

Several PLFAs had isotopic signatures that varied widely between sites, demonstrating
differences in fractionation and/ or source isotopic signatures. Fang et al. (2006) demonstrated
that depth (i.e. pressure) can exert an influence upon PLFA fractionation, but at these sites, depth

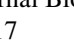



varied only by a small amount (1045 – 1312 m), meaning that this effect should have been quite
limited. The heaviest PLFA $\delta^{13}$C signatures were associated with Hook Ridge sites (e.g. 16:1$\omega$11t
at HR2, $\delta^{13}$C = -8.7 ‰, ~-24 ‰ to -25 ‰ elsewhere). This suggests isotopic differences in the
sources or fractionation by the metabolic pathways used to synthesise these FAs. However,
bacterial fractionation of organic matter can have substantial variation in $\delta^{13}$C signatures,
depending upon variability in the composition and quality (e.g. C: N ratios) of the source (Macko
& Estep 1984) and growth of the organism (Fang et al. 2006), which makes it difficult to elucidate
the specific nature of the differences in substrates between sites.

*Siboglinum* isotopic data demonstrates that methanotrophy was probably occurring at the off
axis sites (Supplementary Figure 1), and depleted PLFA isotopic signatures (e.g. 19:1$\omega$8 - $\delta^{13}$C: -
56.6 ‰; Table 3) provide further suggestion of methanotrophy amongst free-living sedimentary
bacteria. Chemotrophic bacterial sequences, such as *Blastopirellula* (Schlesner 2015) or
*Rhodopirellula* (Bondoso et al. 2014) were found at all sites in relatively high abundance,
suggesting widespread and active chemosynthesis, though the lack of a particularly dominant
bacterial group associated with chemosynthetic activity suggested that the supply of
chemosynthetic OM was likely relatively limited. It remains difficult however to determine
which PLFAs these bacterial lineages may be have been synthesising.

Some PLFAs also had marked differences in $\delta^{13}$C signatures, even where there was strong
compositional similarity between sites (i.e. the non-vent sites). This suggested that either there
were differences in the isotopic values of inorganic or organic matter sources or different
bacterial metabolic pathways were active. Between the non-vent sites, these PLFAs included
PUFAs and MUFAs (Poly- and Mono-unsaturated fatty acids) such as 18:2$\omega$6, 9 ($\Delta\delta^{13}$C 24.4 ‰)
and 19:1$\omega$8 ($\Delta\delta^{13}$C 19.1 ‰). Differences in PLFA $\delta^{13}$C between Hook Ridge sites also ranged



widely, with the largest differences being associated with PLFAs such as 16:1ω11t (Δδ$^{13}$C
17.2 ‰) and 10-Me-16:0 (Δδ$^{13}$C 11.0 ‰). However, it should be stressed that all PLFAs with
larger δ$^{13}$C differences between sites were comparatively rare and never individually exceeded
5% of total abundance. This provides further evidence of limited chemosynthetic activity at all
sites and is consistent with the presence of bacteria associated with methane and sulphur cycling.
Microbial signatures, whilst supporting the suggestion of chemosynthetic activity, are not
indicative of chemosynthetic OM being the dominant source of organic matter to food webs at
any site (hypothesis four). It is not possible to assess from PLFA data the relative importance of
chemoautotrophic and photosynthetic OM sources, since PLFAs degrade quickly and therefore
surface FA abundances are inevitably underestimated in deep water samples. Abundance of
PLFAs associated with surface production, such as 15:0, 20:5ω3, C22:ω6 (Colaço et al. 2007,
Parrish 2013) were low (max 1.8 %), which is consistent with the expected degradation rates
during sinking. Further, piezophillic bacteria have been shown to synthesise some long chain
PUFAs (20:5ω3 and 22:6ω3), which were previously thought to be algal markers (Fang et al.

505 2006).


4.2. Sioglinids

Both species of sioglinid (*Sclerolinum contortum* from Hook Ridge and *Sioglinum* sp. from the
non-vent sites) appeared to subsist upon chemosynthetically derived organic matter, as
evidenced by their morphology, and also by their strongly $^{15}$N-depleted isotopic signatures (see
values with δ$^{15}$N of < -2 ‰ in Fig. 3). Low δ$^{15}$N signatures have also been observed in other
sioglinids in a range of hydrothermal settings, such as *Riftia pachyptila* at the East Pacific Rise
hard substratum vents (Rau 1981). Diazotrophy has been detected previously in hydrothermal
vents and cold seeps, and has been associated with low δ$^{15}$N values (e.g. Rau, 1981; Desai et al.,



2013; Wu et al., 2014 (Yamanaka et al. 2015). Diazotrophy in various reducing settings has been
found associated with anaerobic oxidation of methane (Dekas et al., 2009), methanotrophy
(Mehta & Baross 2006) and (in a non-marine cave) sulphate reduction (Desai et al. 2013) The
latter is also consistent with the low $\delta^{34}S$ signatures of both siboglinid species (Fig. 3; 4), but
gene expression analysis and/or isotopic tracing would be required to confirm this suggestion.
The low $\delta^{34}S$ may also be explained by assimilation of bacterial sulphide, which also gave rise to
metal sulphides (e.g. pyrite) at the vent sites (Petersen et al. 2004). Alternately, low $\delta^{15}N$
signatures may be explained by endosymbionts conducting dissimilatory nitrate reduction to
ammonium (Naraoka et al. 2008, Liao et al. 2014, Bennett et al. 2015), or strong isotopic
fractionation during utilization of ammonia (Naraoka et al. 2008, Liao et al. 2014, Bennett et al.
2015). Bulk faunal isotopic signatures are inadequate to determine which of these
chemosynthesis-related mechanisms is responsible for *Siboglinum* $\delta^{15}N$ values, which would
require analysis of the functional genes in the *Siboglinum* endosymbionts.

Whichever pathway is dominant, $\delta^{15}N$ values for both species ($\delta^{15}N$ *Sclerolinum* = -5.3 ‰ ± 1.0,
*Siboglinum* = -8.9 ‰ ± 0.8) seem to indicate reliance upon locally fixed $N_2$ (Rau 1981, Dekas et
al. 2009, Dekas et al. 2014, Wu et al. 2014, Yamanaka et al. 2015), rather than utilisation of
organic nitrogen sources within the sediment ($\delta^{15}N$ = 5.7 ‰ ± 0.7). These values were also in
contrast to the rest of the non-chemosynthetic obligate species, which generally had much
heavier $\delta^{15}N$ values. This supports hypothesis three, that the siboglinid species were subsisting
upon chemosynthetic OM, most likely supplied by their endosymbionts.

Carbon isotopic signatures in chemosynthetic primary production depend upon the mode of
fixation and the initial $^{13}C$ of available DIC. *Sclerolinum contortum* $\delta^{13}C$ (-20.5 ‰ ± 1.0 ‰) was
depleted in $\delta^{13}C$ relative to Southern Ocean DIC by around 10 ‰ (Henley et al. 2012, Young et



al. 2013), giving it a signal within the fractionation range of the reverse tricarboxyclic acid cycle

(Yorisue et al. 2012). Regional measurements of surface ocean DIC $\delta^{13}$C have an average isotopic

signature of -10.4 ‰ (Henley et al. 2012, Young et al. 2013) but the concentration and isotopic

composition of DIC can undergo considerable alteration at sedimented vents (Walker et al.

2008). Therefore, without measurements of $\delta^{13}$C in pore fluid DIC, it was not possible to

determine which fixation pathway(s) were being used by *S. contortum* endosymbionts.

Sulphur isotopic signatures in *S. contortum* were very low, and quite variable (-26.7 ‰ ± 3.5 ‰).

*Sclerolinum* endosymbionts may have been utilising sulphide re-dissolved from hydrothermal

precipitates. Mineral sulphide was present at Hook Ridge that ranged between -28.1 ‰ to

+5.1 ‰ (Petersen et al. 2004), consistent with the relatively high $\delta^{34}$S variability in *S. contortum*

(but $\delta^{34}$S measurements were subject to higher error between replicates of standards). These

precipitates at Hook Ridge are thought to originate from a previous period of high-temperature

venting at this site (Klinkhammer et al. 2001). Alternatively, sulphide supplied as a result of

microbial sulphate reduction (Canfield 2001) may have been the primary source of organic

sulphur, similar to that of solemyid bivalves from reducing sediments near a sewage pipe outfall

(mean $\delta^{34}$S ranged -30 ‰ to -20 ‰; Vetter and Fry (1998) and in cold seep settings (Yamanaka

et al. 2015). Sulphate reduction can also be associated with anaerobic oxidation of methane

(Whiticar & Suess 1990, Canfield 2001, Dowell et al. 2016), suggesting that methanotrophic

pathways could also have been important at Hook Ridge. (e.g. abundance of *Methylohalomonas*,

2.1 % – 4.3 % of sequences at all sites). Although endosymbiont composition data were not

available for the Southern Ocean population, *Sclerolinum contortum* is also known from

hydrocarbon seeps in the Gulf of Mexico (Eichinger et al. 2013, Eichinger et al. 2014, Georgieva

et al. 2015) and the Håkon Mosby mud volcano in the Arctic ocean, where *S. contortum* $\delta^{13}$C

ranged between -48.3 ‰ to -34.9 ‰(Gebruk et al. 2003) demonstrating that this species is



capable of occupying several reducing environments and using a range of chemosynthetic
fixation pathways, including sulphide oxidation and methanotrophy (Eichinger et al. 2014,
Georgieva et al. 2015).

*Siboglinum* sp. $\delta^{13}$C values  (mean -41.4 ‰, range -45.7 ‰ to -38.1 ‰, n = 8) corresponded very
closely to published values of thermogenic methane (-43 ‰ to -38 ‰) from the Bransfield Strait
(Whiticar & Suess 1990). This suggested that methanotrophy was the likely carbon source for
this species. Biogenic methane typically has much lower $\delta^{13}$C values (Whiticar 1999, Yamanaka
et al. 2015), indicating a hydrothermal/ thermogenic source of methane in the Bransfield Strait
(Whiticar & Suess 1990). Sources of microbially-mediated methane were also present in the
Bransfield Strait (Whiticar & Suess 1990) but these $\delta^{13}$C values were far lower than any of the
faunal signatures observed here. Sulphur isotopic signatures were also very low in *Siboglinum*
sp. ($\delta^{34}$S  -22.9 ‰, one sample from 15 pooled individuals from the off-axis site), the lowest
measurement of $\delta^{34}$S reported for this genus (Schmaljohann & Flügel 1987, Rodrigues et al.
2013).The low $\delta^{13}$C, $\delta^{15}$N and $\delta^{34}$S signatures of *Siboglinum* sp. suggest that its symbionts may
have included methanotrophs (Thornhill et al. 2008) and diazotrophic/ denitrifying bacteria
(Boetius et al. 2000, Canfield 2001, Dekas et al. 2009). Methanotrophy in *Siboglinum* spp. has
been previously documented at seeps in the NE Pacific (Bernardino & Smith 2010) and
Norwegian margin ($\delta^{13}$C = -78.3 ‰ to -62.2 ‰) (Schmaljohann et al. 1990) and in Atlantic mud
volcanoes ($\delta^{13}$C range -49.8 ‰ to -33.0 ‰) (Rodrigues et al. 2013). Sulphur isotopic signatures
in *Siboglinum* spp. from Atlantic mud volcanoes ranged between -16.8 ‰ to 6.5 ‰ (Rodrigues
et al. 2013) with the lowest value still being 6 ‰ greater than that of Bransfield strait specimens.
Rodrigues et al. (2013) also reported a greater range in $\delta^{15}$N than observed in the Bransfield
siboglinids ($\delta^{15}$N -1.3 ‰ to 12.2 ‰ and -10.2 ‰ to -7.6 ‰ respectively). This suggests that, in
comparison to *Siboglinum* spp. in Atlantic Mud volcanoes, which seemed to be using a mixture



of organic matter sources (Rodrigues et al. 2013), the Bransfield specimens relied much more
heavily upon a single OM source, suggesting considerable trophic plasticity in this genus
worldwide.

Off-vent methanotrophy, using thermogenic methane, potentially illustrates an indirect
dependence upon hydrothermalism (Whiticar & Suess 1990). Sediment methane production is
thought to be accelerated by the heat flux associated with mixing of hydrothermal fluid in
sediment (Whiticar & Suess 1990) and sediment and *Siboglinum* isotopic data suggest that the
footprint of hydrothermal influence may be much larger than previously recognised, giving rise
to transitional environments (Bell et al. 2016a, Levin et al. 2016). Clear contribution of methane-
derived carbon to consumer diets was limited predominately to neotanaids, consistent with the
relatively small population sizes (64 ind. $m^2$– 159 ind. $m^2$) of *Siboglinum* sp. observed in the
Bransfield Strait (Bell et al. 2016b).

4.3. Organic Matter Sources

Pelagic salps, collected from an Agassiz trawl at Hook Ridge (1647m), were presumed to most
closely represent a diet of entirely surface-derived material and were more depleted in $^{13}$C and
more enriched in $^{34}$S than were sediments (Salp $\delta^{13}$C = -27.4 ‰ & $\delta^{34}$S = 20.1; Hook Ridge
sediment $\delta^{13}$C = -26.2 ‰ & $\delta^{34}$S = 14.3) Salp samples were also lighter than the majority of
macrofauna, both at Hook Ridge and the non-vent sites (Fig. 3) and similar to other suspension
feeding fauna in the Bransfield Strait (Elias-Piera et al. 2013).

Sediment bulk organic C ($\delta^{13}$C -25.8 to -26.2) was similar to but nonetheless isotopically heavier
than the salp samples. Sediment PLFA data shows that 20.8 - 29.9 % were attributed to bacteria




(summed contributions of i15:0, ai15:0, 16:1ω5c, i17:0, ai17:0, 17:0, and 18:1ω7; Parrish
(2013)), while only 1.0 - 3.8 % were indicative of algal inputs (summed contributions of 15:0,
20:5ω3, 22:6ω3; Parrish (2013)). Thus, while the C isotopes suggest that sedimentary OM was
dominantly derived from surface photosynthesis, the material deposited in the sediment was
likely strongly reworked by bacterial activity.

This suggests that fauna with more depleted $\delta^{34}S$/ more enriched $\delta^{13}C$ values are likely to have
derived at least a small amount of their diet from chemosynthetic sources (potentially indirectly
through non-selective consumption of detrital OM), both at vents and background regions.
Carbon and sulphur isotopic measurements indicated mixed sources for most consumers
between chemosynthetic OM and surface-derived photosynthetic OM. Sediment OM was likely a
combination of these two sources, making both available to non-specific deposit-feeding fauna
and suggesting that consumption of chemosynthetic OM may even have been incidental in some
cases. The low content of algal biomarkers (particularly at the vent sites) suggests that
phytodetritus was probably quite degraded and thus challenging to detect using short-lived fatty
acids. However, the Bransfield Strait can be subject to substantial export production and it is
probable that surface production contributes much more to seafloor OM than is evident from
the fatty acid composition. Non-vent sediments were more enriched in $^{34}S$ than vent sediments,
an offset that probably resulted from greater availability of lighter sulphur sources such as
sulphide oxidation at Hook Ridge.

Samples of bacterial mat could not be collected during JC55 (Tyler et al. 2011) and without these
endmember measurements, it was not possible to quantitatively model resource partitioning in
the Bransfield Strait using isotope mixing models (Phillips et al. 2014). Bacterial mats from high-
temperature vents in the Southern Ocean had $\delta^{34}S$ values of 0.8 ‰ (Reid et al. 2013) and at



sedimented areas of the Loki's Castle hydrothermal vents in the Arctic Ocean has $\delta^{34}S$ values of
-4.9 ‰ (Bulk sediment; Jaeschke et al. 2014). Therefore it is probable that low faunal $\delta^{34}S$ values
represent a contribution of chemosynthetic OM (from either siboglinid tissue or free-living
bacteria). Inorganic sulphur can also be a source to consumers when sulphide is utilised by free
living bacteria ($\delta^{34}S$ ranged -7.3 ‰ to 5.4 ‰; Erickson et al. (2009)) and, although we could not
analyse the $\delta^{34}S$ of fluid sulphide, sulphide crusts have been found at Hook Ridge and may
provide a proxy for typical isotopic composition ($\delta^{34}S$ -28.1 ‰ to 5.1 ‰; Petersen et al. (2004)).
There were several species (e.g. Tubificid oligochaetes) that had moderately depleted $\delta^{34}S$
signatures, such as *Limnodriloides* sp. ($\delta^{34}S$ 7.6 ‰ at vents, -1.2 ‰ at non-vents, Fig. 4) further
supporting the hypothesis of different trophic positions between vent/ non-vent regions
(hypothesis two). This provides evidence of coupled anaerobic oxidation of methane/ sulphate
reduction but overall, the contribution of $\delta^{34}S$-depleted bacterial production did not seem
widespread (further rejecting hypothesis four).

Without samples of all OM sources we cannot quantitatively assert that faunal utilisation of
chemosynthetic OM was low in the Bransfield Strait. Although isotopic data were consistent with
several OM sources, it seemed unlikely that chemosynthetic OM was a dominant source of OM
to the vast majority of taxa. The apparently limited consumption of chemosynthetic OM
suggested that either it was not widely available (e.g. patchy or low density of endosymbiont-
bearing fauna (Bell et al. 2016b)), or that the ecological stress associated with feeding in areas
of in situ production was a significant deterrent to many species (Bernardino et al. 2012, Levin
et al. 2013).

4.4. A-priori vs. a-posteriori trophic groups



Morphology did not prove to be an accurate predictor of trophic associations, suggesting that
faunal behaviour is potentially more important in determining dietary composition than
morphology (e.g. having/ lacking jaws). Peracarid species that possessed structures adapted to
a motile, carnivorous lifestyle were assigned to a carnivore/ scavenger guild (Bell et al. 2016b)
and were distributed throughout the food web both at vents and background regions, indicating
more diverse feeding strategies than expected. Taxa presumed to be deposit feeders (largely
annelids) also had a surprisingly large range of $\delta^{15}$N values. This may reflect the consumption of
detritus from both 'fresh' and more recycled/ refractory OM sources as observed in other non-
vent sedimented deep-sea habitats (Iken et al. 2001, Reid et al. 2012) or reflect variability in
trophic discrimination related to diet quality (Adams & Sterner 2000). Another possibility is taxa
feeding on foraminifera conducting denitrification. A range of foraminifera have now been
shown to conduct this process, which results in them showing elevated $\delta^{15}$N leading to heavy
$\delta^{15}$N values (Pina-Ochoa et al. 2010, Jeffreys et al. 2015). The result is high $\delta^{15}$N values in taxa
without predatory morphology (e.g. oligochaetes) (Bell et al. 2016a). Tubificid oligochaetes had
higher $\delta^{15}$N values at the vent sites, suggesting that they fed upon more recycled organic matter,
possibly owing to greater microbial activity at vent sites. Bacterial biomass was very variable at
the vent sites (86 mg C m$^{-2}$ – 535 mg C m$^{-2}$, compared with 136 mg C m$^{-2}$ – 197 mg C m$^{-2}$ at non-
vent sites; Table 3) and so it is possible that at Hook Ridge 1 bacterial assemblages could have
had a greater influence upon $\delta^{15}$N of organic matter.

Neotanaids from the off-axis site had the lowest $\delta^{13}$C and $\delta^{15}$N values of any non-siboglinid taxon
(Fig. 5), suggesting a significant contribution of methane-derived carbon. The clustering of the
neotanaids together with endosymbiont-bearing taxa is far more likely to be an artefact of the
cluster linkage method, introduced by consumption of low $\delta^{13}$C methanotrophic sources (e.g.



*Siboglinum* tissue), rather than suggesting symbionts in these fauna (Larsen 2006, Levin et al.

691    2009).


Several taxa (e.g. neotanaids from the off-axis site and ophiuroids at Hook Ridge) had low $\delta^{15}$N
values relative to sediment OM, suggesting preferential consumption of chemosynthetic OM
(Rau 1981, Dekas et al. 2014). In these taxa, it is likely that the widespread, but patchy bacterial
mats or *Sclerolinum* populations at Hook Ridge (Aquilina et al. 2013) were an important source
of organic matter to fauna with low $\delta^{15}$N values (e.g. ophiuroids). Fauna from the non-vent sites
with low $\delta^{15}$N were likely subsisting in part upon siboglinid tissue (*Siboglinum* sp.). There were
no video transects over the off-axis site but footage of the Three Sisters, which was similar in
macrofaunal composition (Bell et al. 2016b), did not reveal bacterial mats (Aquilina et al. 2013),
hence it is unlikely that these were an important resource at non-vent sites.

It is clear that some fauna can exhibit a degree of trophic plasticity, depending upon habitat
(supporting hypothesis two). This is consistent with other SHVs where several taxa (e.g.
*Prionospio* sp. – Polychaeta: Spionidae) had different isotopic signatures, depending upon their
environment (Levin et al. 2009), demonstrating differential patterns in resource utilisation.
Alternatively, there could have been different $\delta^{15}$N baselines between sites, though if these
differences were significant, we argue that it likely that more species would have had significant
differences in tissue $\delta^{15}$N. Conversely, samples of *Aurospio foodbancsia* at both vent and non-
vent sites had broadly similar $\delta^{15}$N values to that of the west Antarctic Peninsula; 8.1 ‰ and
7.9 ‰ respectively, albeit with a higher variability (Mincks et al. 2008). $\delta^{13}$C values of *Aurospio*
were also broadly similar, implying that this species occupied a detritivorous trophic niche,
irrespective of environmental conditions.





4.5. Impact of hydrothermal activity on community trophodynamics

Standard ellipse area was lower at Hook Ridge than at non-vent sites (Table 5), analogous to
trends in macrofaunal diversity and abundance in the Bransfield Strait (Bell et al. 2016b) and
changes in SEA.B along a gradient of methane flux at vent and seep ecosystems in the Guaymas
Basin (Portail et al. 2016). This demonstrates that at community level, ellipse area can be
associated with other macrofaunal assemblage characteristics. This concurrent decline in niche
area and alpha diversity is consistent with the concept that species have finely partitioned niches
and greater total niche area permits higher biodiversity (McClain & Schlacher 2015).
Productivity-diversity relationships, whereby higher productivity sustains higher diversity,
have also been suggested for deep-sea ecosystems (McClain & Schlacher 2015, Woolley et al.
2016) but in the absence of measurements of in situ organic matter fixation rates at Hook Ridge,
it is unclear whether such relationships exist in the Bransfield Strait. Sediment organic carbon
content was similar between Hook Ridge 1 and non-vent sites but was slightly lower at Hook
Ridge 2 (Bell et al. 2016b), which is not consistent with variation in niche area. The decline in
alpha diversity and niche area is consistent with the influence of disturbance gradients created
by hydrothermalism that result in an impoverished community (McClain & Schlacher 2015, Bell
et al. 2016b). We suggest that, in the Bransfield Strait, the environmental toxicity at SHVs (from
differences in temperature and porewater chemistry) causes a concomitant decline in both
trophic and species diversity (Bell et al. 2016b), in spite of the potential for increased localised
production. However, we acknowledge that, owing to the high small-scale habitat heterogeneity
apparent from video imagery over the vent area, that it is likely that the contribution of
chemosynthetic organic matter varies widely over 10s of metres at Hook Ridge.



Community-based trophic metrics (Layman et al. 2007) indicated that, although measures of
dispersion within sites were relatively similar between vents and background areas (Table 5),
trophic diversity, particularly in terms of range of carbon sources (dCr) and total hull area (TA)
were higher at background sites. It was expected that trophic diversity would be greater at Hook
Ridge but the greater dCr at non-vent sites (owing to the methanotrophic source) meant that the
isotopic niches at these sites were larger. Range in nitrogen values (dNr) was also greater at non-
vents, driven by the more heavily depleted $\delta^{15}$N values of *Siboglinum* sp. It is of course debatable
whether this assemblage isotopic niche really corresponds to the assemblage's actualised
trophic niche and, although the niche space was smaller at the vent sites, the potential for
different trophic strategies was still potentially greater than at non-vent sites. Differences in
eccentricity are more heavily influenced by the spread of all isotopes used to construct the niche
space (where E = 0 corresponds to an equal influence of both carbon and nitrogen) whereas
theta (the angle of the long axis) determines which, if any, isotope is most influential in
determining ellipse characteristics (Reid et al. 2016). For the non-vent sites, the dominant
isotope was carbon, owing to the relatively light $\delta^{13}$C of methanotrophic source utilised by
*Siboglinum*. Some sites, particularly the Axe, had several fauna with heavy $\delta^{13}$C values (Fig. 6),
which could be explained by either contamination from marine carbonate (~0 ‰), as specimens
were not acidified, or a diet that included a heavier source of carbon, such as sea ice algae
(Henley et al. 2012).


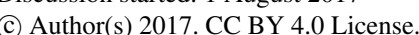


Section 5. Conclusions

In this study, we demonstrate the influence of sediment-hosted hydrothermal venting upon
trophodynamics and microbial populations. Low activity vent microbiota were more similar to
the non-vent site than to high activity populations, illustrating the effect of ecological gradients
upon deep-sea microbial diversity. Despite widespread bacterial mats, and populations of vent-
endemic macrofauna, utilisation of chemosynthetic OM amongst non-specialist macro- and
megafauna seemed relatively low, with a concomitant decline in trophic diversity with
increasing hydrothermal activity. Morphology was also not indicative of trophic relationships,
demonstrating the effects of differential resource availability and behaviour. We suggest that,
because these sedimented hydrothermal vents are insufficiently active to host large populations
of vent-endemic megafauna, the transfer of chemosynthetic organic matter into the metazoan
food web is likely to be more limited than in other similar environments.



6. Acknowledgements

JBB was funded by a NERC PhD Studentship (NE/L501542/1). This work was funded by the
NERC ChEsSo consortium (Chemosynthetically-driven Ecosystems South of the Polar Front,
NERC Grant NE/DOI249X/I). Elemental analyses were funded by the NERC Life Sciences Mass
Spectrometry Facility (Proposal no. EK234-13/14). We thank Barry Thornton and the James
Hutton Laboratory, Aberdeen for processing the PLFA samples. We also thank Will Goodall-
Copestake for assistance in processing the 16S sequence data. We are grateful to the Master and
Crew of RRS *James Cook* cruise 055 for technical support and the Cruise Principal Scientific
Officer Professor Paul Tyler.

7. Ethics Statement

In accordance with the Antarctic Act (1994) and the Antarctic Regulations (1995), necessary
permits (S5-4/2010) were acquired from the South Georgia and South Sandwich Islands
Government.

8. Author contributions

Conceived and designed the sampling programme: WDKR, DAP, AGG, CJS & CW. Sample
laboratory preparation and isotopic analyses: JBB, JN & CJS. Microbial sequencing: DAP.
Statistical analyses: JBB. Produced figures: JBB. Wrote the paper: JBB, CW & WDKR, with
contributions and comments from all other authors.



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

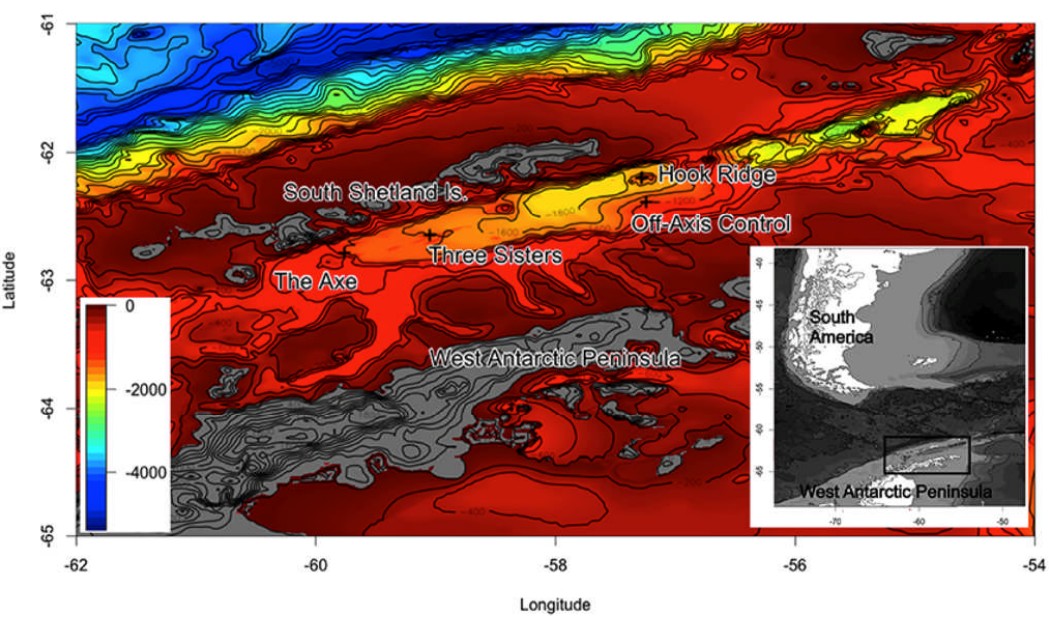


Figure 1 – Sampling sites (after Bell et al. 2016b)




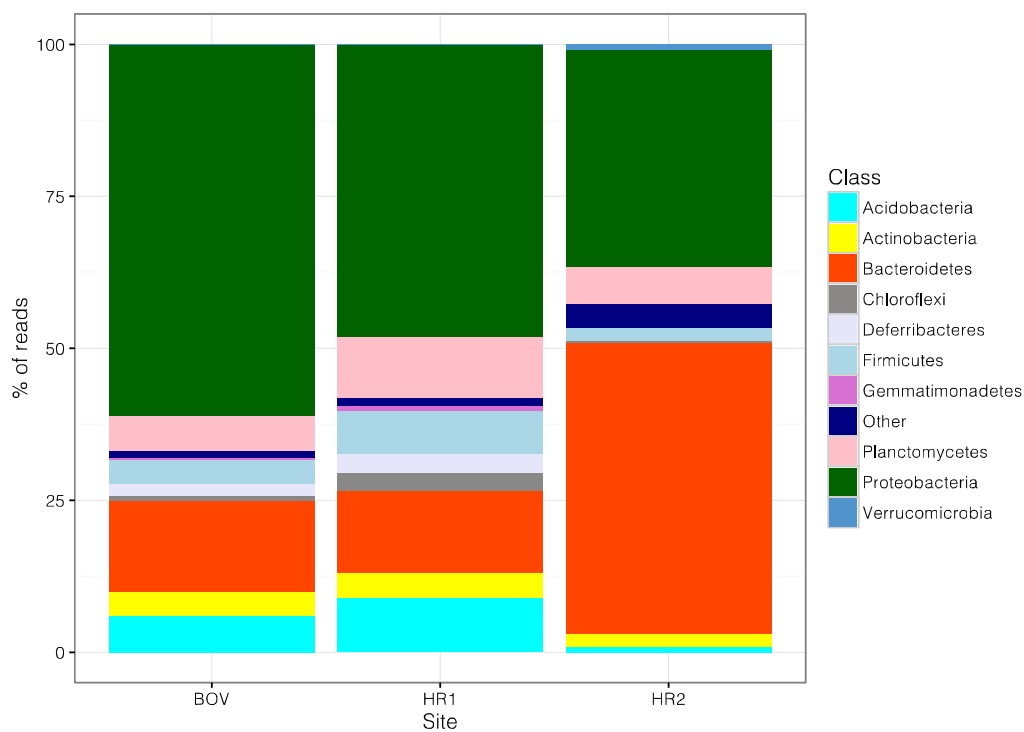


Figure 2 – Microbial composition (classes) at the off-vent/ off-axis site (BOV) and the two Hook
Ridge sites (HR1 and HR2). Archaea excluded from figure as they only accounted for 0.008 % of
reads at HR2 and were not found elsewhere.





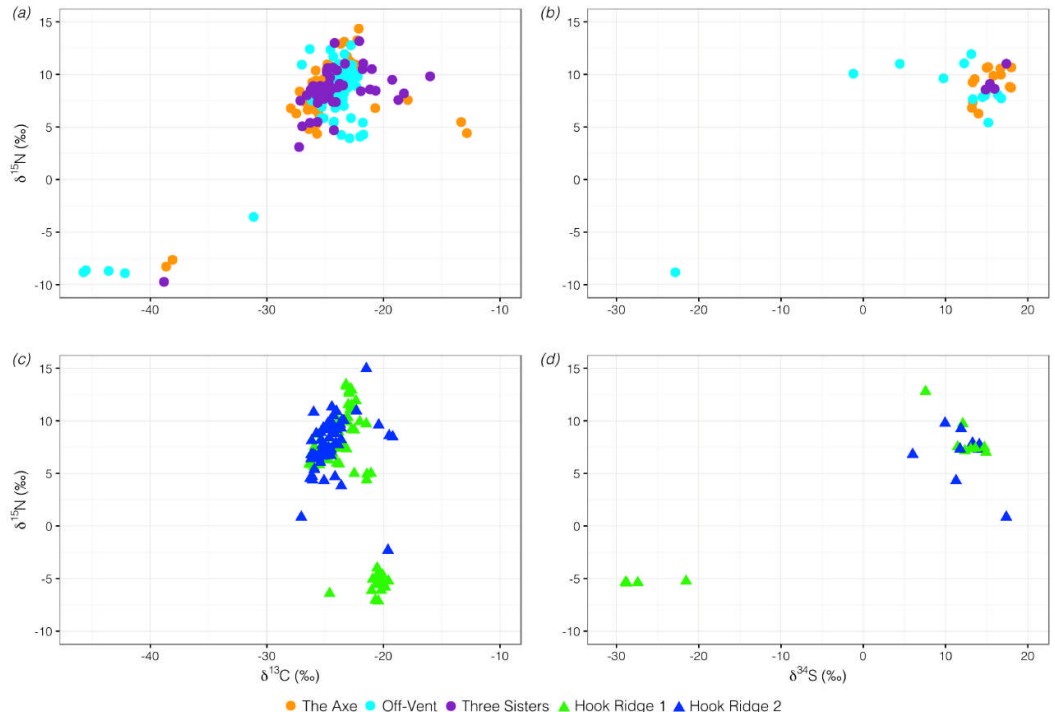


Figure 3 – Carbon-Nitrogen and Sulphur-Nitrogen biplots for bulk isotopic signatures of benthos,
separated into non-vent (top) and vent sites (bottom). Excepting one value from the off-vent site
(for a peracarid species), all values with $\delta^{15}$N of < 0 were siboglinid species (*Sclerolinum*
*contortum* from the vent sites and *Siboglinum* spp. from the non-vent sites).





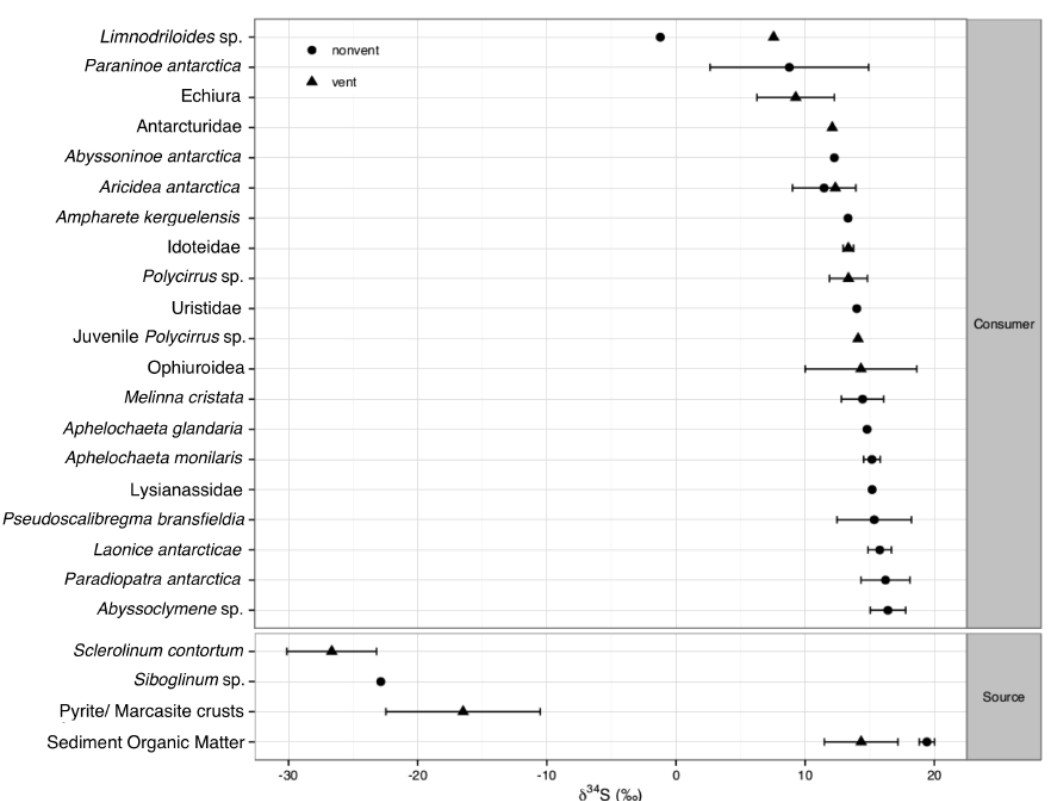


Figure 4 – Plot of δ³⁴S measurements by discriminated by species and habitat (vent/ non-vent ±
1 s.d.). Data for δ³⁴S in crusts from Petersen et al. (2004)






Figure 5– Biplot of CN isotopic data from species sampled both at vents and non-vent

background regions. Mean ± standard deviation, X-Y scales vary







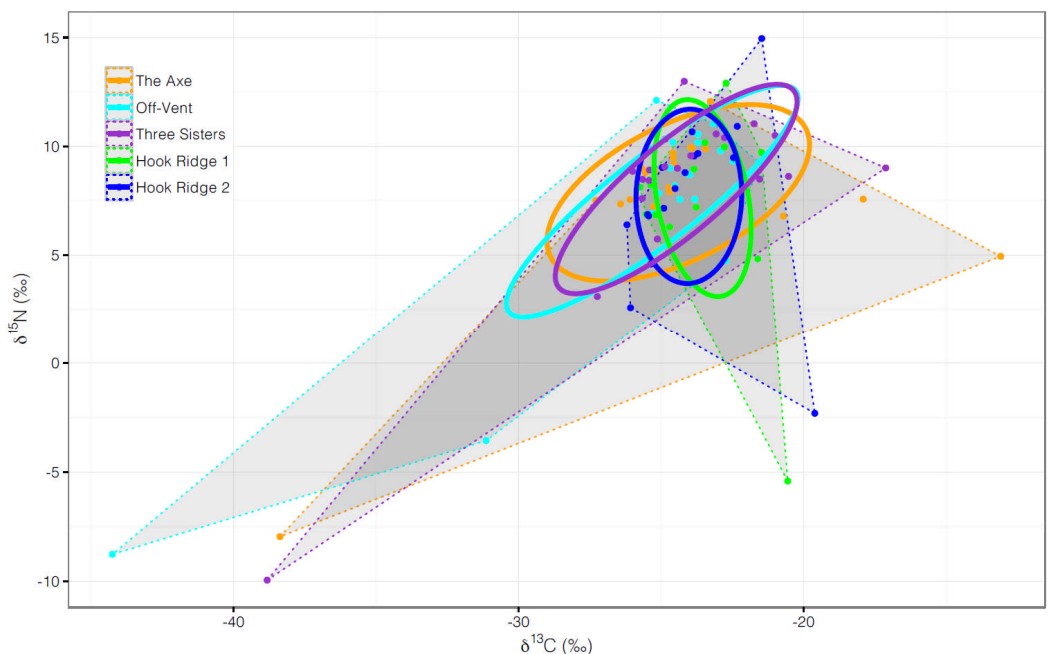


Figure 6 – Faunal isotopic signatures (mean per species), grouped by site with total area (shaded
area marked by dotted lines) and sample-size corrected standard elliptical area (solid lines)





11. Tables

| Site | Depth (m) | Hydrothermally active? | References |
|---|---|---|---|
| The Axe (AXE) | 1024 | No | (Dählmann et al. 2001, Klinkhammer et al. 2001, Sahling et al. 2005, Aquilina et al. 2013, Aquilina et al. 2014, Bell et al. 2016b) |
| Off-Vent (BOV) | 1150 | No | |
| Three Sisters (TS) | 1311 | No | |
| Hook Ridge 1 (HR1) | 1174 | Low activity (9 cm yr$^{-1}$) | |
| Hook Ridge 2 (HR2) | 1054 | High Activity (34 cm yr$^{-1}$) | |


Table 1 – Site descriptions and associated references

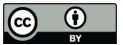


| Isotope | Species | Idoteidae | *Polycirrus* sp. | *Aphelochaeta glandaria* | Phyllodocida sp. |
|---|---|---|---|---|---|
| | Treatment | 0.1M HCl | 0.1M HCl | 0.1M HCl | 1.0M HCl |
| $\delta^{13}C$ (‰) | Difference in mean | 1.6 | 0.2 | 0.4 | 0.9 |
| | σ untreated | 0.7 | 0.3 | 0.2 | 0.5 |
| | σ treated | 0.7 | 0.3 | 0.2 | 0.2 |
| | Population range | 2.9 | 3.0 | 2.7 | - |
| $\delta^{15}N$ (‰) | Difference in mean | 0.9 | 0.2 | 0.1 | 0.9 |
| | σ untreated | 0.2 | 0.3 | 0.2 | 0.4 |
| | σ treated | 1.0 | 0.2 | 0.2 | 0.3 |
| | Population range | 3.4 | 4.6 | 5.8 | - |
| $\delta^{34}S$ (‰) | Difference in mean | - | - | 0.4 | 1.1 |
| | σ untreated | - | - | 0.4 | 0.8 |
| | σ treated | - | - | 0.7 | 1.4 |
| | Population range | - | - | 2.3 | - |


Table 2 – Differences in isotopic values and standard deviation (σ) of ethanol preserved fauna
sampled during JC55 in response to acid treatment, compared with population ranges of
untreated samples. Phyllodocida sp. was a single large specimen, used only as part of
preliminary experiments. Data rounded to 1 d.p. to account for measurement error.

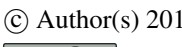




| PLFA | Bransfield Off-Vent | | | Three Sisters | | |
| | nM g⁻¹ | % | δ¹³C (‰) | nM g⁻¹ | % | δ¹³C (‰) |
|---|---|---|---|---|---|---|
| i14:0 | 0.03 | 0.12 | -22.0 | 0.02 | 0.09 | -28.0 |
| 14:0 | 0.80 | 3.04 | -31.2 | 0.83 | 3.43 | -30.9 |
| i15:0 | 0.76 | 2.89 | -28.6 | 0.76 | 3.13 | -28.1 |
| a15:0 | 1.06 | 4.03 | -28.4 | 1.06 | 4.39 | -27.7 |
| 15:0 | 0.30 | 1.13 | -29.3 | 0.19 | 0.77 | -29.8 |
| i16:1 | 0.11 | 0.44 | -31.4 | 0.02 | 0.10 | -20.3 |
| 16:1w11c | 0.00 | 0.00 | n.d. | 0.06 | 0.24 | -23.1 |
| i16:0 | 0.34 | 1.30 | -28.5 | 0.30 | 1.24 | -27.8 |
| 16:1w11t | 0.78 | 2.98 | -24.4 | 0.66 | 2.75 | -25.0 |
| 16:1w7c | 3.98 | 15.19 | -28.9 | 3.37 | 13.95 | -28.1 |
| 16:1w5c | 1.12 | 4.27 | -34.1 | 0.96 | 3.99 | -34.0 |
| 16:0 | 4.29 | 16.37 | -31.1 | 3.80 | 15.73 | -30.0 |
| br17:0 | 0.00 | 0.00 | n.d. | 0.00 | 0.00 | n.d. |
| 10-Me-16:0 | 0.46 | 1.77 | -28.5 | 0.45 | 1.87 | -29.1 |
| i17:0 | 0.08 | 0.32 | -33.2 | 0.20 | 0.84 | -29.8 |
| a17:0 | 0.25 | 0.97 | -31.9 | 0.21 | 0.87 | -31.3 |
| 12-Me-16:0 | 0.25 | 0.94 | -32.9 | 0.21 | 0.86 | -31.6 |
| 17:1w8c | 0.13 | 0.50 | -34.1 | 0.11 | 0.44 | -31.3 |
| 17:0cy | 0.33 | 1.26 | -36.2 | 0.27 | 1.10 | -32.8 |
| 17:0 | 0.15 | 0.56 | -40.0 | 0.08 | 0.33 | -50.4 |
| 10-Me-17:0 | 0.00 | 0.00 | n.d. | 0.00 | 0.00 | n.d. |
| 18:3w6,8,13 | 0.67 | 2.55 | -34.6 | 0.69 | 2.87 | -33.8 |
| 18:2w6,9 | 0.12 | 0.46 | -27.8 | 0.09 | 0.36 | -52.2 |
| 18:1w9 | 1.13 | 4.30 | -30.0 | 1.33 | 5.50 | -29.9 |
| 18:1w7 | 4.42 | 16.85 | -29.0 | 3.84 | 15.91 | -29.1 |
| 18:1w(10 or 11) | 2.33 | 8.88 | -30.1 | 2.26 | 9.36 | -29.9 |
| 18:0 | 0.66 | 2.50 | -30.6 | 0.54 | 2.22 | -30.6 |
| 19:1w6 | 0.03 | 0.12 | -23.5 | 0.03 | 0.12 | -30.1 |
| 10-Me-18:0 | 0.00 | 0.00 | n.d. | 0.00 | 0.00 | n.d. |
| 19:1w8 | 0.11 | 0.42 | -56.6 | 0.17 | 0.69 | -37.5 |
| 19:0cy | 0.20 | 0.77 | -35.6 | 0.20 | 0.83 | -34.8 |
| 20:4(n-6) | 0.14 | 0.55 | -40.0 | 0.20 | 0.83 | -34.1 |
| 20:5(n-3) | 0.41 | 1.57 | -38.0 | 0.30 | 1.23 | -39.3 |
| 20:1(n-9) | 0.42 | 1.60 | -31.5 | 0.41 | 1.71 | -33.7 |
| 22:6(n-3) | 0.22 | 0.83 | -34.1 | 0.43 | 1.77 | -30.0 |
| 22:1(n-9) | 0.10 | 0.39 | -31.3 | 0.10 | 0.41 | -29.9 |
| 24:1(n-9) | 0.03 | 0.12 | -28.7 | 0.02 | 0.07 | -29.7 |
| | | | | | | |
| Total | 26.23 | | | 24.15 | | |
| Average | 0.71 | | -30.5 | 0.65 | | -30.1 |



|                    |       | mg C m$^{-2}$ | δ$^{13}$C (‰) |       | mg C m$^{-2}$ | δ$^{13}$C (‰) |
|--------------------|-------|---------------|---------------|-------|---------------|---------------|
| Bacterial Biomass  |       | 134.50        | -26.8         |       | 197.12        | -26.4         |



|              | Hook Ridge 1 |              | Hook Ridge 2 |       |              | Range        |
|--------------|--------------|--------------|--------------|-------|--------------|--------------|
|              |              | δ$^{13}$C    |              |       | δ$^{13}$C    | δ$^{13}$C    |
| PLFA         | nM g$^{-1}$  | (‰)          | nM g$^{-1}$  | %     | (‰)          | (‰)          |
| i14:0        | 0.03         | -15.7        | 0.10         | 0.80  | -28.8        | -13.1        |
| 14:0         | 0.80         | -32.7        | 0.80         | 6.40  | -29.6        | -3.1         |
| i15:0        | 0.76         | -29.7        | 0.40         | 3.20  | -28.1        | -1.7         |
| a15:0        | 1.06         | -29.1        | 0.90         | 7.20  | -28.9        | -1.4         |
| 15:0         | 0.30         | -29.0        | 0.30         | 2.40  | -28.3        | -1.5         |
| i16:1        | 0.11         | -27.6        | 0.00         | 0.00  | n.d.         | -11.1        |
| 16:1ω11c     | 0.00         | -17.4        | 0.00         | 0.00  | n.d.         | -5.7         |
| i16:0        | 0.34         | -29.4        | 0.20         | 1.60  | -28.8        | -1.6         |
| 16:1ω11t     | 0.78         | -25.8        | 0.30         | 2.40  | -8.7         | -17.2        |
| 16:1ω7c      | 3.98         | -29.2        | 2.50         | 20.00 | -22.9        | -6.3         |
| 16:1ω5c      | 1.12         | -31.2        | 0.30         | 2.40  | -24.3        | -9.7         |
| 16:0         | 4.29         | -31.8        | 3.30         | 26.40 | -29.3        | -2.5         |
| br17:0       | 0.00         | -22.9        | 0.00         | 0.00  | -15.8        | -7.2         |
| 10-Me-16:0   | 0.46         | -30.3        | 0.20         | 1.60  | -41.3        | -12.8        |
| i17:0        | 0.08         | n.d.         | 0.00         | 0.00  | n.d.         | -3.4         |
| a17:0        | 0.25         | -29.0        | 0.20         | 1.60  | -28.6        | -3.4         |
| 12-Me-16:0   | 0.25         | -28.6        | 0.10         | 0.80  | -28.2        | -4.7         |
| 17:1ω8c      | 0.13         | -27.1        | 0.10         | 0.80  | -27.2        | -6.9         |
| 17:0cy       | 0.33         | -32.3        | 0.20         | 1.60  | -27.7        | -8.5         |
| 17:0         | 0.15         | -40.0        | 0.20         | 1.60  | -30.8        | -19.6        |
| 10-Me-17:0   | 0.00         | -35.0        | 0.00         | 0.00  | n.d.         | 0.00         |
| 18:3ω6,8,13  | 0.67         | -31.2        | 0.50         | 4.00  | -29.0        | -5.6         |
| 18:2ω6,9     | 0.12         | -30.0        | 0.30         | 2.40  | -26.7        | -25.5        |
| 18:1ω9       | 1.13         | -29.6        | 0.40         | 3.20  | -25.6        | -4.4         |
| 18:1w7       | 4.42         | -29.9        | 0.60         | 4.80  | -24.7        | -5.1         |
| 18:1ω(10 or 11) | 2.33      | -31.9        | 0.00         | 1.60  | n.d.         | -2.0         |
| 18:0         | 0.66         | -29.4        | 0.30         | 0.00  | -29.9        | -1.2         |
| 19:1ω6       | 0.03         | -26.2        | 0.00         | 2.40  | n.d.         | -6.6         |

6000



| | mg C m⁻² | δ¹³C (‰) | | mg C m⁻² | δ¹³C (‰) | |
|---|---|---|---|---|---|---|
| 10-Me-18:0 | 0.00 | -25.4 | 0.00 | 0.00 | n.d. | 0.0 |
| 19:1ω8 | 0.11 | -41.2 | 0.00 | 0.00 | n.d. | -19.1 |
| 19:0cy | 0.20 | -30.5 | 0.10 | 0.00 | -28.7 | -6.9 |
| 20:4(n-6) | 0.14 | n.d. | 0.00 | 0.80 | n.d. | -5.9 |
| 20:5(n-3) | 0.41 | n.d. | 0.00 | 0.00 | n.d. | -1.3 |
| 20:1(n-9) | 0.42 | n.d. | 0.00 | 0.00 | n.d. | -2.2 |
| 22:6(n-3) | 0.22 | n.d. | 0.00 | 0.00 | n.d. | -4.2 |
| 22:1(n-9) | 0.10 | n.d. | 0.00 | 0.00 | n.d. | -1.4 |
| 24:1(n-9) | 0.03 | n.d. | 0.00 | 0.00 | n.d. | -1.0 |
| | | | | | | |
| Total | 26.23 | | 12.30 | | | |
| Average | 0.71 | -30.3 | 0.33 | | -26.9 | |
| | | δ¹³C (‰) | | mg C m⁻² | δ¹³C (‰) | |
| | mg C m⁻² | | | | | |
| Bacterial Biomass | 534.55 | -26.6 | | 85.45 | -23.1 | |


Table 3 – PLFA profiles from freeze-dried sediment (nM per g dry sediment). PLFA names relate
to standard notation (i = iso; a = anti-iso; first number = number of carbon atoms in chain; ω =
double bond; Me = methyl group). N.P. = Not present in sample. Total PLFA δ¹³C measurements
weighted by concentration Bulk bacterial δ¹³C estimated from average conversion factor of
3.7 ‰ (Boschker & Middelburg 2002). No data = n.d. Range measurements may be subject to
rounding error. N. B. Table split to conform to submission portal requirements.





| Isotope | Vents ‰ (± S.D.) | Non-Vent ‰ (± S.D.) | Different? (T-Test, df = 3) |
|---|---|---|---|
| $\delta^{13}C$ | -26.2 (± 0.4) | -25.8 (± 0.3) | No |
| $\delta^{15}N$ | 5.7 (± 0.7) | 5.0 (± 0.3) | No |
| $\delta^{34}S$ | 14.3 (± 2.9) | 19.4 (± 0.6) | Yes (T = 3.49, p < 0.05) |


Table 4 – Mean isotopic signatures of sediment organic matter.






| Site | Ellipse | | | | Θ | E | CD | Nearest Neighbour Distance | |
|---|---|---|---|---|---|---|---|---|---|
| | SEAc (‰²) | SEA.B (‰²) | Cred. (95% ± ‰²) | TA (‰²) | | | | Mean | S.D. |
| The Axe | 49.3 | 45.0 | 19.9 | 161.6 | 0.67 | 0.85 | 3.59 | 1.76 | 4.17 |
| Off-Vent | 39.8 | 36.5 | 16.8 | 139.1 | 0.81 | 0.97 | 4.34 | 2.13 | 3.88 |
| Three Sisters | 35.5 | 32.6 | 14.7 | 110.2 | 0.86 | 0.95 | 3.85 | 1.93 | 3.78 |
| Hook Ridge 1 | 23.1 | 20.7 | 11.2 | 42.6 | -1.43 | 0.94 | 3.30 | 1.64 | 2.60 |
| Hook Ridge 2 | 23.4 | 21.1 | 10.7 | 61.8 | 1.55 | 0.89 | 3.17 | 1.52 | 2.03 |
| **Mean** | | | | | | | | | |
| **Non-Vent** | **41.5** | **38.0** | **17.2** | **137.0** | **0.78** | **0.92** | **3.93** | **1.94** | **3.94** |
| **Vent** | **23.2** | **20.9** | **11.0** | **52.2** | **0.10** | **0.91** | **3.23** | **1.58** | **2.31** |


| Site | Centroid | | | | |
|---|---|---|---|---|---|
| | δ¹³C (‰) | δ¹⁵N (‰) | δ³⁴S (‰) | dNr (‰) | dCr (‰) |
| The Axe | -24.4 | 7.9 | | 20.0 | 25.3 |
| Off-Vent | -25.3 | 7.5 | 8.1 | 20.9 | 22.7 |
| Three Sisters | -24.5 | 8.0 | | 22.9 | 21.7 |
| Hook Ridge 1 | -23.5 | 7.6 | 5.4 | 18.3 | 5.2 |
| Hook Ridge 2 | -24.0 | 7.7 | | 17.3 | 6.6 |



| Mean | | | | | |
|---|---|---|---|---|---|
| Non-Vent | -24.7 | 7.8 | | 21.3 | 23.2 |
| Vent | -23.8 | 7.7 | | 17.8 | 5.9 |


Table 5 – Ellipse Area & Layman Metrics of benthos by site. SEAc = Sample-sized corrected
standard elliptical area; SEA.B = Bayesian estimate of standard elliptical area; TA = Total hull
area; E = Eccentricity; dNr = Nitrogen range; dCr = Carbon range; dSr = Sulphur range; CD =
Centroid distance. Note: dSR reported only for Hook Ridge 1 and the off-vent site since $\delta^{34}$S
values of siboglinids were only measured from these sites; hence dSr at other sites would be a
considerable underestimate. As $\delta^{34}$S values were comparatively under-representative, these
values were not used in calculation of any other metric. Data rounded to 1 d.p. N. B. Table split
to conform to submission portal requirements.