# Peer review of "Hydrothermal activity lowers trophic diversity in Antarctic sedimented hydrothermal vents"

_Biogeosciences, 2017_

## Referee Comment (RC1) · Anonymous Referee #1 · 18 Aug 2017

This paper reports the food ecology of macrofauna and possible food source, that is microbial communities in the sediments obtained from hydrothermal and non-hydrothermal areas in Southern Ocean based on CNS isotope compositions and molecular phylogenetic and PLFA analyses. This study is a sequel to the previous paper about macrofaunal ecology of the same area written by the same authors. The conclusions led by the analytical results are almost adequate, but the discussion is quite lengthy and is not straightforward. It can be shortened and simplified.

Individual points to be improved

P14 lines 297-304: I can not find any associated tables and figures mentioned in the texts.

P17 lines 358-362: What is the "four clusters"? And which figures and tables are related to this paragraph?

Food ecology of siboglinid species (chemosynthesis-based or not) must be discussed before the section 4.1 (difference of microbial assemblages and those biomass among each site). And this discussion is related to the hypothesis 1, right?

P21 lines 444-445: Long chain fatty acids originated in land plants are derived as form of triglyceride (wax). They are not PLFA.

P24 lines 545-546: S. consortium endosymbiont use only DIC in pore fluid? I think the symbiont use mainly DIC in bottom water. Because the siboglinid worm is not infauna, right?

P25 lines 548-: The previous studies (Klinkhammer et al., 2001, Aquilina et al., 2013) indicated presence of hydrogen sulfide in the sediments. The H2S concentrations were increasing with depth and sulfate concentrations in the pore fluids were decreasing with depth. It possibly suggests that active microbial sulfate reduction is occurred below seafloor. Therefore, very low sulfur isotopic signature of the siboglinid worms mainly associated with microbial sulfide. Mineral sulfide dissolution is not necessary (but hydrothermal fluid input can not be ignored).

P26 lines 585-587: If the siboglinid worms harbored thioautotrophic endosymbiont, sulfur isotopic ratios of the worm reflect the ratio of hydrogen sulfide. Therefore, the difference of 6 ‰ is meaningless.

P27 line 610: "Salp samples were also lighter than. . .", what is lighter? Carbon isotopic ratio?

P28 lines 633-635: The sediment samples using this study were not removed pore fluids sulfate before analysis. So the sulfur isotope data include 34S rich sulfate originated in pore fluid. In addition, organic sulfur originated in photosynthetic organic matter, which also enriched in 34S, is main component of the sedimentary sulfur. Possible

another sulfur source in the sediment is bacterial and/or hydrothermal sulfide (mainly form of pyrite). Why you mentioned only sulfide oxidation?

P30 lines 686-687: methane is not contained nitrogen. Lowest d15N values can not explain only methane.

Other minor points The term "vent" means an opening that allows gas or liquid to pass out. This study is not discussed hydrothermal vent, but hydrothermal activity (it include venting and shimmering and any other ascending fluid). So, I think the author change the term "vent" into "activity" or "system (or area)".

P2 line 20: "among the least studied.." change to "one of the least studied.."

P14 line 288: I can not find "Flavobacteriia" in tables and figures. It should change to "Bacteroidetes".

"Sulphate reducing bacteria" should change to "sulphate-reducing bacteria".

---

## Referee Comment (RC2) · Anonymous Referee #2 · 9 Sep 2017

I was asked to review the paper "Hydrothermal activity lowers trophic diversity in Antarctic sedimented hydrothermal vents" by James B. Bell, William D. K. Reid, David A. Pearce, Adrian G. Glover, Christopher J. Sweeting, Jason Newton, and Clare Woulds.

I find the paper well in the scope and focus of the Journal and the scientiphic work carried out is surely of high quality. Data are abundant, protocols and procedures of sampling and analysis are adequate and the techniques used are rilevant.

This manuscript is the natural continuation of the previous paper written by the same author pool on the same site and it completes the previous findings. Although the results are interesting and well supported, I find the manuscript very long and often difficult to follow and wearisome. In particular, the discussion in not straightforward,

lenghty and, in my opinoin, it lacks a strong structure. Too often it winds and results tortuous, forcing hte reading to go back in order to find the "fil rouge" to follow. I would warmly suggest to shorten the whole manuscript and in particular the discussion. In my opinon, the discussion should follow fewer clear, strong and important points, starting from hypothesis moving through the results and finally offering the conclusions and the answers to the main scintific questions.

I would suggest to insert some more tables and figures that better present the results: for instance, the data reported in the paragraph 3.1 lines 297-304 are not listed in any table nor well represented in a figure and this is a pity. Since the scientific and technical effort behind this work is huge, I would suggest to try to valorize it more by showing all the numbers and cite tables and figures more in the text than in the supplementary material.

I have only one strictly scientific comment to make: in lines 686-687 the authors say "Neotanaids from the off-axis site had the lowest $\delta$13C and $\delta$15N values of any non-siboglinid taxon (Fig. 5), suggesting a significant contribution of methane-derived carbon". This sentence may be misleading: while I agree that a lower $\delta$13C may suggest the methabolism of methane-derived carbon, I fail to see how a lower $\delta$15N signature may support this hypothesis, since methane does not contain N. It would be better to reformulate the sentence.

---

## Author Comment (AC1) · 4 Oct 2017

This paper reports the food ecology of macrofauna and possible food source, that is microbial communities in the sediments obtained from hydrothermal and on hydrothermal areas in Southern Ocean based on CNS isotope compositions and molecular phylogenetic and PLFA analyses. This study is a sequel to the previous paper about macrofaunal ecology of the same area written by the same authors.

The conclusions led by the analytical results are almost adequate, but the discussion is quite lengthy and is not straightforward. It can be shortened and simplified.
**Owing to the multiple lines of evidence, the discussion as it stands is lengthy. We agree with this appraisal and will ensure that the revised manuscript focuses more strongly upon the hypotheses in order to improve readability, as suggested by the other reviewer.**

Individual points to be improved:

P14 lines 297-304: I cannot find any associated tables and figures mentioned in the texts.
**We will add reference to figure 1 (microbial composition data).**

P17 lines 358-362: What is the "four clusters"? And which figures and tables are related to this paragraph?
**The "four clusters" refer to the Euclidean distance matrix used to delineate sub-structure in the isotopic data. Figure 5 and supplement 3 are related to this paragraph, which we refer to in the text. We will amend the text to improve clarity. We will also expand discussion of the cluster results but will keep in mind not to increase the length of the overall discussion.**

Food ecology of sibogilinid species (chemosynthesis-based or not) must be discussed before the section 4.1 (difference of microbial assemblages and those biomass among each site). And this discussion is related to the hypothesis 1, right?
**Discussion concerning food sources of the sibogilinids does relate to hypothesis 1, but we would prefer to re-order the hypotheses. We will then have the results and discussion section following a structure of microbial signatures, through individual faunal signatures up to community metrics.**

P21 lines 444-445: Long chain fatty acids originated in land plants are derived as form of triglyceride (wax). They are not PLFA.
**We will correct instances where fatty acids are mislabeled as PLFAs**

P24 lines 545-546: S. consortium endosymbiont use only DIC in pore fluid? I think the symbiont use mainly DIC in bottom water. Because the siboglinid worm is not infauna, right?
***Sclerolinum contortum* is an infaunal species so our discussion DIC sources is accurate. We will amend the text to improve clarity of this point.**

P25 lines 548-: The previous studies (Klinkhammer et al., 2001, Aquilina et al., 2013) indicated presence of hydrogen sulfide in the sediments. The H2S concentrations were increasing with depth and sulfate concentrations in the pore fluids were decreasing with depth. It possibly suggests that active microbial sulfate reduction is occurred below seafloor. Therefore, very low sulfur isotopic signature of the siboglinid worms mainly associated with microbial sulfide. Mineral sulfide dissolution is not necessary (but hydrothermal fluid input can not be ignored).
**The reviewer's suggestion is potentially supported by our data and is valid. We will amend the relevant text to include this possibility.**

P26 lines 585-587: If the siboglinid worms harbored thioautotrophic endosymbiont, sulfur isotopic ratios of the worm reflect the ratio of hydrogen sulfide. Therefore, the difference of 6 ‰ is meaningless.

**The 6‰ highlights that the Bransfield Strait are lower than siboglinid worms found in other locations and puts the Bransfield Strait worms in a wider ecological context. The sulphur isotopic ratios of mineralized sulphide in the Bransfield Strait (Petersen et al. 2004) vary widely and their signatures do overlap with those of the siboglinids presented here. However, the reviewer's comment does not consider the role of trophic fractionation, which can easily account for large differences in isotopic signature in sulphur metabolism.**

P27 line 610: "Salp samples were also lighter than…", what is lighter? Carbon isotopic ratio?

**The Salps had a lighter $d^{13}C$ value than values of sedimentary organic carbon. We will make sure this comparison is added to the text.**

P28 lines 633-635: The sediment samples using this study were not removed pore fluids sulfate before analysis. So the sulfur isotope data include 34S rich sulfate originated in pore fluid. In addition, organic sulfur originated in photosynthetic organic matter, which also enriched in 34S, is main component of the sedimentary sulfur. Possible another sulfur source in the sediment is bacterial and/or hydrothermal sulfide (mainly form of pyrite). Why you mentioned only sulfide oxidation?

**Sediment samples were drained of pore fluids, freeze-dried and then rinsed in de-ionised water, thus traces of sulphate should have been removed as far as possible. Photosynthetic organic sulphur likely remains the major component as the reviewer correctly points out but the vent areas still have lower $d^{34}S$ values, indicating a source of isotopically light organic (or possibly mineral) sulphur, which we attribute to hydrothermal processes. We will amend text as necessary to improve clarity of this point.**

P30 lines 686-687: methane is not contained nitrogen. Lowest d15N values cannot explain only methane.

**The text will be amended to remove reference to $d^{15}N$ values.**

Other minor points The term "vent" means an opening that allows gas or liquid to pass out. This study is not discussed hydrothermal vent, but hydrothermal activity (it include venting and shimmering and any other ascending fluid). So, I think the author change the term "vent" into "activity" or "system (or area)".

**We will change the term "vent" into "activity" as requested by the reviewer. This will better capture the phenomena we are investigating because the manuscript is looking at the ascending fluids derived from sub-surface hydrothermal processes influence microbial and metazoan communities.**

P2 line 20: "among the least studied.." change to "one of the least studied.."
**Text will be amended as recommended by the reviewer.**

P14 line 288: I cannot find "Flavobacteriia" in tables and figures. It should change to "Bacteroidetes". "Sulphate reducing bacteria" should change to "sulphate-reducing bacteria".
**Text will be amended in both cases, as recommended by the reviewer.**

---

## Author Comment (AC2) · 4 Oct 2017

I was asked to review the paper "Hydrothermal activity lowers trophic diversity in Antarctic sedimented hydrothermal vents" by James B. Bell, William D. K. Reid, David A. Pearce, Adrian G. Glover, Christopher J. Sweeting, Jason Newton, and Clare Woulds.

I find the paper well in the scope and focus of the Journal and the scientific work carried out is surely of high quality. Data are abundant, protocols and procedures of sampling and analysis are adequate and the techniques used are relevant.

This manuscript is the natural continuation of the previous paper written by the same author pool on the same site and it completes the previous findings. Although the results are interesting and well supported, I find the manuscript very long and often difficult to follow and wearisome. In particular, the discussion in not straightforward, lengthy and, in my opinion, it lacks a strong structure. Too often it winds and results tortuous, forcing the reading to go back in order to find the "fil rouge" to follow. I would warmly suggest to shorten the whole manuscript and in particular the discussion. In my opinion, the discussion should follow fewer clear, strong and important points, starting from hypothesis moving through the results and finally offering the conclusions and the answers to the main scientific questions.
**This point has been fairly raised by both reviewers. We agree that the discussion could be structured better and will address this point in the revision, through a clearer focus upon the hypotheses presented and reduction in overall length.**

I would suggest to insert some more tables and figures that better present the results: for instance, the data reported in the paragraph 3.1 lines 297-304 are not listed in any table nor well represented in a figure and this is a pity. Since the scientific and technical effort behind this work is huge, I would suggest trying to valorize it more by showing all the numbers and cite tables and figures more in the text than in the supplementary material.
**We will add more figures and tables where appropriate, keeping in mind limits relating to the journal style.**

I have only one strictly scientific comment to make: in lines 686-687 the authors say "Neotanaids from the off-axis site had the lowest $\delta^{13}C$ and $\delta^{15}N$ values of any non-siboglinid taxon (Fig. 5), suggesting a significant contribution of methane-derived carbon". This sentence may be misleading: while I agree that a lower $\delta^{13}C$ may suggest the metabolism of methane-derived carbon, I fail to see how a lower $\delta^{15}N$ signature may support this hypothesis, since methane does not contain N. It would be better to reformulate the sentence.
**We will remove reference to Nitrogen isotopic values as suggested by the reviewer, so as to avoid confusion.**

---

## Author Response (AR1)

James Bell
University of Leeds
Leeds
LS2 9JT
UK

2nd October 2017

**Author Response to BG-2017-288 reviews**

Dear Biogeosciences Editors,

We are pleased to see that our manuscript 'Hydrothermal Activity lowers Trophic Diversity in Antarctic Sedimented Hydrothermal Vents' was well received by both reviewers, building upon the improvements made during the previous round of reviews. Many of the comments are technical issues, which are simple to rectify, and we will be pleased to make these changes, pending the editor's decision.

We agree that in places, the structure of the manuscript could be improved, particularly so for the discussion and we will focus the revisions upon improving the flow and readability of the manuscript as outlined below.

We propose to make the following changes (in bold, following each of the reviewer comments), and thank both the reviewers for their considered and helpful comments.

Thank you for your continued consideration of this article.

**Anonymous Referee #1**

This paper reports the food ecology of macrofauna and possible food source, that is microbial communities in the sediments obtained from hydrothermal and on hydrothermal areas in Southern Ocean based on CNS isotope compositions and molecular phylogenetic and PLFA analyses. This study is a sequel to the previous paper about macrofaunal ecology of the same area written by the same authors.

The conclusions led by the analytical results are almost adequate, but the discussion is quite lengthy and is not straightforward. It can be shortened and simplified.
**Owing to the multiple lines of evidence, the discussion as it stands is lengthy. We agree with this appraisal and have made efforts to ensure that the revised manuscript focuses more strongly upon the hypotheses presented in order to improve readability, as suggested by both reviewers.**

Individual points to be improved:

P14 lines 297-304: I cannot find any associated tables and figures mentioned in the texts.
**We have added more references to figure 1 (microbial composition data) in section 3.1.**

P17 lines 358-362: What is the "four clusters"? And which figures and tables are related to this paragraph?
**The "four clusters" refer to the Euclidean distance matrix used to delineate sub-structure in the isotopic data. Figure 5 and supplement 3 are related to this paragraph, which we refer to in the text. We have amended the text to improve clarity (~Line 365). We have also expanded discussion of the cluster results (~Line 680) keeping in mind the feedback to reduce the length of the overall discussion.**

Food ecology of siboglinid species (chemosynthesis-based or not) must be discussed before the section
4.1 (difference of microbial assemblages and those biomass among each site). And this discussion is
related to the hypothesis 1, right?
**Discussion concerning food sources of the siboglinids does relate to hypothesis 1, but we would**
**prefer to re-order the hypotheses (~Line 117-19). We now have the hypotheses, results and**
**discussion section following a structure of microbial signatures, through individual faunal**
**signatures up to community metrics.**
P21 lines 444-445: Long chain fatty acids originated in land plants are derived as form of triglyceride
(wax). They are not PLFA.
**We have corrected several instances instances where other fatty acids are mislabeled as PLFAs or**
**the entire FA suite has been referred to as PLFAs.**
P24 lines 545-546: S. consortium endosymbiont use only DIC in pore fluid? I think the symbiont use
mainly DIC in bottom water. Because the siboglinid worm is not infauna, right?
*Sclerolinum contortum* **is an infaunal species so our discussion DIC sources is accurate. We have**
**amended the text to improve clarity of this point (Section 4.2).**
P25 lines 548-: The previous studies (Klinkhammer et al., 2001, Aquilina et al., 2013) indicated presence
of hydrogen sulfide in the sediments. The H2S concentrations were increasing with depth and sulfate
concentrations in the pore fluids were decreasing with depth. It possibly suggests that active microbial
sulfate reduction is occurred below seafloor. Therefore, very low sulfur isotopic signature of the
siboglinid worms mainly associated with microbial sulfide. Mineral sulfide dissolution is not necessary
(but hydrothermal fluid input can not be ignored).
**The reviewer's suggestion is potentially supported by our data and is valid. We have amended the**
**relevant text to include this possibility (~Line 599).**
P26 lines 585-587: If the siboglinid worms harbored thioautotrophic endosymbiont, sulfur isotopic ratios
of the worm reflect the ratio of hydrogen sulfide. Therefore, the difference of 6 ‰ is meaningless.
**The 6‰ highlights that the Bransfield Strait are lower than siboglinid worms found in other**
**locations and puts the Bransfield Strait worms in a wider ecological context. The sulphur isotopic**
**ratios of mineralized sulphide in the Bransfield Strait (Petersen et al. 2004) vary widely and their**
**signatures do overlap with those of the siboglinids presented here. However, the reviewer's**
**comment does not consider the role of trophic fractionation, which can easily account for large**
**differences in isotopic signature in sulphur metabolism. We address the amendments more fully**
**later in response to the editor's additional comment.**
P27 line 610: "Salp samples were also lighter than...", what is lighter? Carbon isotopic ratio?
**The Salps had a lighter d$^{13}$C value than values of macrofauna and sedimentary organic carbon. We**
**have amended the text to improve clarity of this point (~Line 660).**
P28 lines 633-635: The sediment samples using this study were not removed pore fluids sulfate before
analysis. So the sulfur isotope data include 34S rich sulfate originated in pore fluid. In addition, organic
sulfur originated in photosynthetic organic matter, which also enriched in 34S, is main component of the
sedimentary sulfur. Possible another sulfur source in the sediment is bacterial and/or hydrothermal
sulfide (mainly form of pyrite). Why you mentioned only sulfide oxidation?
**Sediment samples were drained of pore fluids, freeze-dried and then rinsed in de-ionised water,**
**thus traces of sulphate should have been removed as far as possible. Photosynthetic organic**
**sulphur likely remains the major component as the reviewer correctly points out but the vent**
**areas still have lower d$^{34}$S values, indicating a source of isotopically light organic (or possibly**
**mineral) sulphur, which we attribute to hydrothermal processes. We have amended the text in**
**section 4.3 (~lines 688 – 702) to improve clarity of this point.**
P30 lines 686-687: methane is not contained nitrogen. Lowest d15N values cannot explain only methane.

**The text will be amended to remove reference to d¹⁵N values.**
Other minor points The term "vent" means an opening that allows gas or liquid to pass out. This study is
not discussed hydrothermal vent, but hydrothermal activity (it include venting and shimmering and any
other ascending fluid). So, I think the author change the term "vent" into "activity" or "system (or area)".
**We have changed the term "vent" into "activity" or "hydrothermal" as requested by the reviewer.**
**This will better capture the phenomena we are investigating because the manuscript is looking at**
**the ascending fluids derived from sub-surface hydrothermal processes influence microbial and**
**metazoan communities.**
P2 line 20: "among the least studied.." change to "one of the least studied.."
**Text has been amended as recommended by the reviewer.**
P14 line 288: I cannot find "Flavobacteriia" in tables and figures. It should change to "Bacteroidetes".
**Bacterial genera have been added to a new table (see also Reviewer 2: comment 2).**
"Sulphate reducing bacteria" should change to "sulphate-reducing bacteria".
**Text amended as suggested.**
                        **Anonymous Referee #2**
I was asked to review the paper "Hydrothermal activity lowers trophic diversity in Antarctic sedimented
hydrothermal vents" by James B. Bell, William D. K. Reid, David A. Pearce, Adrian G. Glover, Christopher
J. Sweeting, Jason Newton, and Clare Woulds.
I find the paper well in the scope and focus of the Journal and the scientific work carried out is surely of
high quality. Data are abundant, protocols and procedures of sampling and analysis are adequate and the
techniques used are relevant. This manuscript is the natural continuation of the previous paper written
by the same author pool on the same site and it completes the previous findings.
Although the results are interesting and well supported, I find the manuscript very long and often difficult
to follow and wearisome. In particular, the discussion in not straightforward, lengthy and, in my opinion,
it lacks a strong structure. Too often it winds and results tortuous, forcing the reading to go back in order
to find the "fil rouge" to follow. I would warmly suggest to shorten the whole manuscript and in particular
the discussion. In my opinion, the discussion should follow fewer clear, strong and important points,
starting from hypothesis moving through the results and finally offering the conclusions and the answers
to the main scientific questions.
**This point has been fairly raised by both reviewers. We agree that the discussion could be**
**structured better and shortened in length and have addressed this point in the revision, through**
**a clearer focus upon the hypotheses presented and reduction in overall length.**
I would suggest to insert some more tables and figures that better present the results: for instance, the
data reported in the paragraph 3.1 lines 297-304 are not listed in any table nor well represented in a
figure and this is a pity. Since the scientific and technical effort behind this work is huge, I would suggest
trying to valorize it more by showing all the numbers and cite tables and figures more in the text than in
the supplementary material.
**We have added a table detailing the major microbial genera sequenced from each site,**
**complementing figure 2, as recommended by the reviewer. The present manuscript comprises 6**
**figures and 6 tables and is supplied with 3 additional supplementary figures. We believe that this**
**covers the breadth of the key points and, with respect to the comments raised concerning the**
**length of the manuscript, would recommend that no additional figures/ tables are necessary. We**
**would however welcome the Associate Editor's opinion on this point.**

I have only one strictly scientific comment to make: in lines 686-687 the authors say "Neotanaids from
the off-axis site had the lowest d13C and d15N values of any non-siboglinid taxon (Fig. 5), suggesting a
significant contribution of methane-derived carbon". This sentence may be misleading: while I agree that
a lower d13C may suggest the metabolism of methane-derived carbon, I fail to see how a lower d15N
signature may support this hypothesis, since methane does not contain N. It would be better to
reformulate the sentence.
**We have removed reference to nitrogen isotopic values as suggested by the reviewer, so as to**
**avoid confusion.**
**Associate Editor Comment**
**Received: 17 October 2017**
Thank you for your series of answers and discussions and also making revision in response to reviewers
comments. Most of revisions are satisfied for us except for one point.
As Reviewer 1 made a comment,
P26 lines 585-587: If the siboglinid worms harbored thioautotrophic endosymbiont, sulfur isotopic ratios
of the worm reflect the ratio of hydrogen sulfide. Therefore, the difference of 6 ‰ is meaningless.
You have answered as follows.
The 6‰ highlights that the Bransfield Strait are lower than siboglinid worms found in other locations
and puts the Bransfield Strait worms in a wider ecological context. The sulphur isotopic ratios of
mineralized sulphide in the Bransfield Strait (Petersen et al. 2004) vary widely and their signatures do
overlap with those of the siboglinids presented here. However, the reviewer's comment does not
consider the role of trophic fractionation, which can easily account for large differences in isotopic
signature in sulphur metabolism.
Trophic fractionation of sulfur isotope is small as similar to carbon isotopes. this is well known
phenomena as that has already described in Fry, 1983; 1988 and Peterson and Howarth, 1988. If you do
not agree on their opinion, you should refer following papers.
Fry B. (1983) Fishery Bulletin 81: 789-801
Fry B. (1988) Limnology and Oceanography 33: 1182-1190
Peterson B.J. and Howarth R.W. (1987) Limnology and Oceanography 32: 1195-1213
Then, you are requested to change following sentence to suggested one.
(your revision)
Sulphur isotopic signatures in Siboglinum spp. from Atlantic mud volcanoes ranged between -16.8 ‰ to
6.5 ‰ (Rodrigues et al. 2013) with the lowest value still being 6 ‰ greater than that of Bransfield strait
specimens.
(recommended correction)
Sulphur isotopic signatures in Siboglinum spp. from Atlantic mud volcanoes ranged between -16.8 ‰ to
6.5 ‰ (Rodrigues et al. 2013), whereas the lowest value of this study was still 6 ‰ lower. It reflects the
relative lower sulphur isotopic ratios of hydrogen sulphide yielding in the study sites (that is also
suggesting that bacterial sulphide is main source of hydrogen sulfide).
**We have amended the sentence as suggested by the editor but would like to clarify that, whilst**
**sulphur does not fractionate substantially between faunal trophic levels, there are a number of**
**metabolic processes that are involved in sulphur cycling, which can result in substantial shifts in**
**sulphur isotopic composition (e.g. Canfield DE (2001) Isotope fractionation by natural**

**populations of sulfate-reducing bacteria. Geochimica Et Cosmochimica Acta 65:1117-1124 or**
**Habicht KS, Canfield DE (1997) Sulfur isotope fractionation during bacterial sulfate reduction in**
**organic-rich sediments. Geochimica Et Cosmochimica Acta 61:5351-5361).**
End of comments
**Once again, we thank both the anonymous reviewers, Professor Kitazato, and the editorial staff**
**for their handling of this manuscript and we look forward to concluding this submission.**
**Regards,**

**Dr James Bell (on behalf of the authors)**

[revised manuscript text omitted]